# Fetal and neonatal hematopoietic progenitors are functionally and transcriptionally resistant to *Flt3*-ITD mutations

**Shaina N Porter[1], Andrew S Cluster[1], Wei Yang[2], Kelsey A Busken[1], Riddhi M Patel[1], Jiyeon Ryoo[1], Jeffrey A Magee[1,2]***

[1]Division of Pediatric Hematology and Oncology, Department of Pediatrics, Washington University School of Medicine, St. Louis, United States; [2]Department of Genetics, Washington University School of Medicine, St. Louis, United States

**Abstract** The *FLT3* Internal Tandem Duplication (*FLT3^{ITD}*) mutation is common in adult acute myeloid leukemia (AML) but rare in early childhood AML. It is not clear why this difference occurs. Here we show that *Flt3^{ITD}* and cooperating *Flt3^{ITD}/Runx1* mutations cause hematopoietic stem cell depletion and myeloid progenitor expansion during adult but not fetal stages of murine development. In adult progenitors, FLT3^{ITD} simultaneously induces self-renewal and myeloid commitment programs via STAT5-dependent and STAT5-independent mechanisms, respectively. While FLT3^{ITD} can activate STAT5 signal transduction prior to birth, this signaling does not alter gene expression until hematopoietic progenitors transition from fetal to adult transcriptional states. Cooperative interactions between *Flt3^{ITD}* and *Runx1* mutations are also blunted in fetal/neonatal progenitors. Fetal/neonatal progenitors may therefore be protected from leukemic transformation because they are not competent to express FLT3^{ITD} target genes. Changes in the transcriptional states of developing hematopoietic progenitors may generally shape the mutation spectra of human leukemias.

*For correspondence: Magee_J@kids.wustl.edu

**Competing interests:** The authors declare that no competing interests exist.

## Introduction

Acute myeloid leukemia (AML) can occur at any stage of life yet the mutations that cause AML differ between childhood and adulthood, especially when one compares young children to adults (*Chaudhury et al., 2015*). For example, *MLL* translocations and *GATA1* mutations are common in infant and early childhood AML but rare in adult AML (*Andersson et al., 2015*; *Horton et al., 2013*; *Pine et al., 2007*). Mutations in *FLT3*, *NPM1*, *DNMT3A*, *TET2* and *IDH1* are all common in adult AML but rare in infant and early childhood AML (*Cancer Genome Atlas Research Network, 2013*; *Ho et al., 2011*; *Liang et al., 2013*; *Zwaan et al., 2003*). The genetic differences between pediatric and adult AML are not absolute, but they reflect a more general phenomenon in leukemia biology – leukemias in infants, young children, older children and adults have different genetic and epigenetic landscapes, different mechanisms of transformation and different clinical courses (*Downing and Shannon, 2002*). Efforts to interpret AML genomes and translate the information into useful therapies will need to account for the influences of age and developmental context on leukemia cell biology. This will require a better understanding of how normal developmental programs shape the process of leukemogenesis.

The mutations that cause AML are thought to accrue first in pre-leukemic hematopoietic stem cells (HSCs) or committed hematopoietic progenitor cells (HPCs) (*Jan et al., 2012*; *Welch et al.,*

**eLife digest** Leukemias are a group of blood cancers that usually arise when immature blood cells gain one or more tumor-promoting genetic mutations. However, for reasons that are not clear, the mutations that cause leukemia are different in children and adults. For example, a mutation called $FLT3^{ITD}$ occurs relatively often in adult leukemia but is rare in infant leukemia. This raises the question of whether the blood cells of fetuses and babies are somehow protected from the effects of the mutation.

Porter et al. have now compared the effects of the $FLT3^{ITD}$ mutation in blood cells from adult and fetal mice. In adult mice, the $FLT3^{ITD}$ mutation caused immature blood cells to turn different genes on and off. By contrast, the mutation had no effect on the activity of these genes in fetal mice. Furthermore, only the adult mutant cells showed changes that indicated the early stages of leukemia: the mutant blood cells of fetuses developed as normal. Porter et al. therefore concluded that the immature blood cells of fetuses are protected from the $FLT3^{ITD}$ mutation.

To understand why fetal and adult blood cells respond differently to the $FLT3^{ITD}$ mutation, further experiments are needed to investigate how various genes regulate normal blood cell development. In addition, understanding why adult blood cells react to the $FLT3^{ITD}$ mutation might, in the future, lead to better treatment options for leukemia.

*2012*), and several properties of these cells change between fetal and adult stages of life: (1) Fetal HSCs divide frequently and retain their self-renewal capacity through cumulative division cycles (*Pietras and Passegué, 2013*). In contrast, adult HSCs are usually quiescent, and self-renewal capacity declines with cumulative divisions (*Foudi et al., 2009*; *Pietras and Passegué, 2013*; *Wilson et al., 2008*). (2) Fetal and adult HSCs have distinct self-renewal mechanisms. For example, *Sox17* is required for fetal, but not adult, HSC self-renewal (*Kim et al., 2007*). *Etv6*, *Ash1l, Mll* and *Pten* are all required for adult, but not fetal, HSC self-renewal (*Hock et al., 2004*; *Jones et al., 2015*; *Jude et al., 2007*; *Magee et al., 2012*). (3) Fetal and adult HSCs give rise to committed progenitors with distinct epigenetic landscapes (*Huang et al., 2016*; *Xu et al., 2012*) and distinct lineage biases (*Benz et al., 2012*; *Copley et al., 2013*; *Yuan et al., 2012*). These observations raise the question of whether mutations can have age-specific effects on gene expression, self-renewal, differentiation and ultimately leukemogenesis. If so, competence for transformation may be a heterochronic property of HSCs and HPCs, and this may explain why pediatric and adult leukemias have different mutations.

The *FLT3 Internal Tandem Duplication* ($FLT3^{ITD}$) is an example of an AML driver mutation that occurs more commonly in adults than in young children (30–40% of adult AML, 5–10% of AML in children <10 years old, <1% of infant AML) (*Meshinchi et al., 2006*). $FLT3^{ITD}$ encodes a constitutively active tyrosine kinase receptor that has been shown to activate the STAT5, MAP-kinase (MAPK), PI3-kinase (PI3K), STAT3 and NF-κB signal transduction pathways in various contexts (*Choudhary et al., 2007*; *Gerloff et al., 2015*; *Radomska et al., 2006*). Mice with a targeted $Flt3^{ITD}$ mutation develop myeloproliferative neoplasms (MPN) (*Lee et al., 2007*; *Li et al., 2008*), and several other mutations (e.g. *Npm1*, *Tet2* and *Runx1* mutations) cooperate with $Flt3^{ITD}$ to drive AML in mice much as in humans (*Mead et al., 2013*; *Mupo et al., 2013*; *Rau et al., 2014*; *Shih et al., 2015*). In the absence of cooperating mutations, $Flt3^{ITD}$ drives adult HSCs into cycle and depletes the HSC pool (*Chu et al., 2012*). This may explain why $FLT3^{ITD}$ mutations occur late in the clonal evolution of human AML — adult HSCs must first acquire mutations that preserve (or ectopically establish) self-renewal capacity in pre-leukemic progenitors—but it also raises the question of why fetal/neonatal HSCs, which have an inherently high self-renewal capacity (*He et al., 2009*), do not give rise to $FLT3^{ITD}$ positive AML more often than is observed.

To better understand how developmental context shapes myeloid leukemogenesis, we characterized the effects of $Flt3^{ITD}$ on HSC self-renewal, myelopoiesis, signal transduction and gene expression at several stages of pre- and post-natal development. $Flt3^{ITD}$ did not cause HSC depletion or myeloid progenitor expansion until after birth. This was true even in the presence of a cooperating *Runx1* loss-of-function mutation. The $FLT3^{ITD}$ protein phosphorylated STAT5 during both pre- and

post-natal stages of development while it hyper-activated the MAPK pathway only after birth. To our surprise, MAPK inhibition failed to rescue HSC depletion and myeloid progenitor expansion in adult *Flt3[ITD]* mice, and *Stat5a/b* deletion greatly exacerbated these phenotypes. FLT3[ITD] target genes, including STAT5 targets, were not induced in fetal HSCs or HPCs despite pre-natal STAT5 phosphorylation. Instead, FLT3[ITD] target gene activation coincided with a normal transition from fetal to adult gene expression that was evident by two weeks after birth. These temporal changes in FLT3[ITD] target gene expression were observed even in the setting of a cooperating *Runx1* mutation.

Our data establish a crucial role for developmental context in the pathogenesis of *FLT3[ITD]*-driven AML. Fetal and neonatal progenitors are protected from transformation because they are not competent to express FLT3[ITD] target genes. This likely explains why *FLT3[ITD]* mutations are more common in adults than young children, and it may reflect a more general role for developmental programming in leukemia pathogenesis.

## Results

### *Flt3[ITD]* does not deplete fetal HSCs

Since *FLT3[ITD]* occurs more commonly in adult AML patients than in young children, we hypothesized that it might have age-specific effects on self-renewal and myelopoiesis. We first tested whether *Flt3* expression changes with age. We measured *Flt3* transcript expression in CD150[+]-CD48[-]Lineage[-]Sca1[+]c-kit[+] HSCs and CD48[+]Lineage[-]Sca1[+]c-kit[+] HPCs from 8–10 week old adult and embryonic day (E)14.5 fetal mice by quantitative RT-PCR (qRT-PCR). *Flt3* was more highly expressed in HPCs than in HSCs at both ages (*Figure 1A*), consistent with prior studies (*Buza-Vidas et al., 2011*), but its expression did not change with age in either cell population (*Figure 1A*). Flow cytometry confirmed that the FLT3 protein is expressed in both fetal and adult progenitors (*Figure 1B*).

Since *Flt3[ITD]* has previously been shown to deplete adult HSCs (*Chu et al., 2012*), we tested whether the mutation has a similar effect on fetal HSC numbers. We measured HSC numbers in 8–10 week old adult bone marrow and E14.5 fetal livers from wild type, *Flt3[ITD/+]* and *Flt3[ITD/ITD]* mice. Adult *Flt3[ITD/+]* mice had ~50% fewer HSCs than wild type littermates, consistent with prior studies, and *Flt3[ITD/ITD]* mice had a near-complete loss of phenotypic HSCs (*Figure 1C*). HSC depletion in the bone marrow was not accompanied by extramedullary expansion of HSCs in the spleen (*Figure 1E*), in contrast to other leukemogenic mutations (e.g. *Pten* deletion) that cause depletion of bone marrow HSCs but marked expansion of the spleen HSC population (*Magee et al., 2012*; *Porter et al., 2016*). Unlike adult mice, *Flt3[ITD/+]* and *Flt3[ITD/ITD]* fetal mice had similar numbers of HSCs as wild type littermates (*Figure 1D*). The *Flt3[ITD]* mutation therefore depletes adult, but not fetal HSCs.

We next tested whether fetal *Flt3[ITD/ITD]* HSCs are functionally impaired. We performed limiting dilution transplantation assays with either 8–10 week old adult bone marrow cells (600,000, 100,000, 50,000 or 10,000 CD45.2 donor cells competed with 300,000 CD45.1 adult bone marrow cells) or E14.5 fetal liver cells (100,000, 50,000 or 10,000 CD45.2 donor cells competed with 300,000 CD45.1 adult bone marrow cells). Two independent experiments were performed, and fetal and adult donor cells were transplanted at the same time in each experiment. Multi-lineage reconstitution was assessed every 4 weeks for 16 weeks following the transplants, and functional HSC frequencies were calculated by Extreme Limiting Dilution Analysis (*Hu and Smyth, 2009*). Adult *Flt3[ITD/ITD]* bone marrow had significantly fewer functional HSCs than adult wild type bone marrow (p<0.00001, *Figure 1F*). In contrast, wild type and *Flt3[ITD/ITD]* fetal livers had similar HSC frequencies (*Figure 1G*).

Our findings raised the question of whether fetal *Flt3[ITD/ITD]* HSCs can mature and become depleted after transplantation into adult recipient mice. To test this, we measured donor HSC chimerism in primary recipients of 100,000 wild type and *Flt3[ITD/ITD]* fetal liver cells (from *Figure 1G*). Donor *Flt3[ITD/ITD]* HSCs were significantly depleted in primary recipient mice, but wild type competitor HSCs were not (*Figure 1H*). Overall donor bone marrow chimerism was not significantly different between recipients of wild type and *Flt3[ITD/ITD]* fetal liver cells (*Figure 1I*). Secondary transplants confirmed depletion of *Flt3[ITD/ITD]* HSCs in the marrow of primary recipients (*Figure 1J,K*). Thus, fetal *Flt3[ITD/ITD]* HSCs are functional, but they lose repopulating activity after transplantation into adult recipient mice.

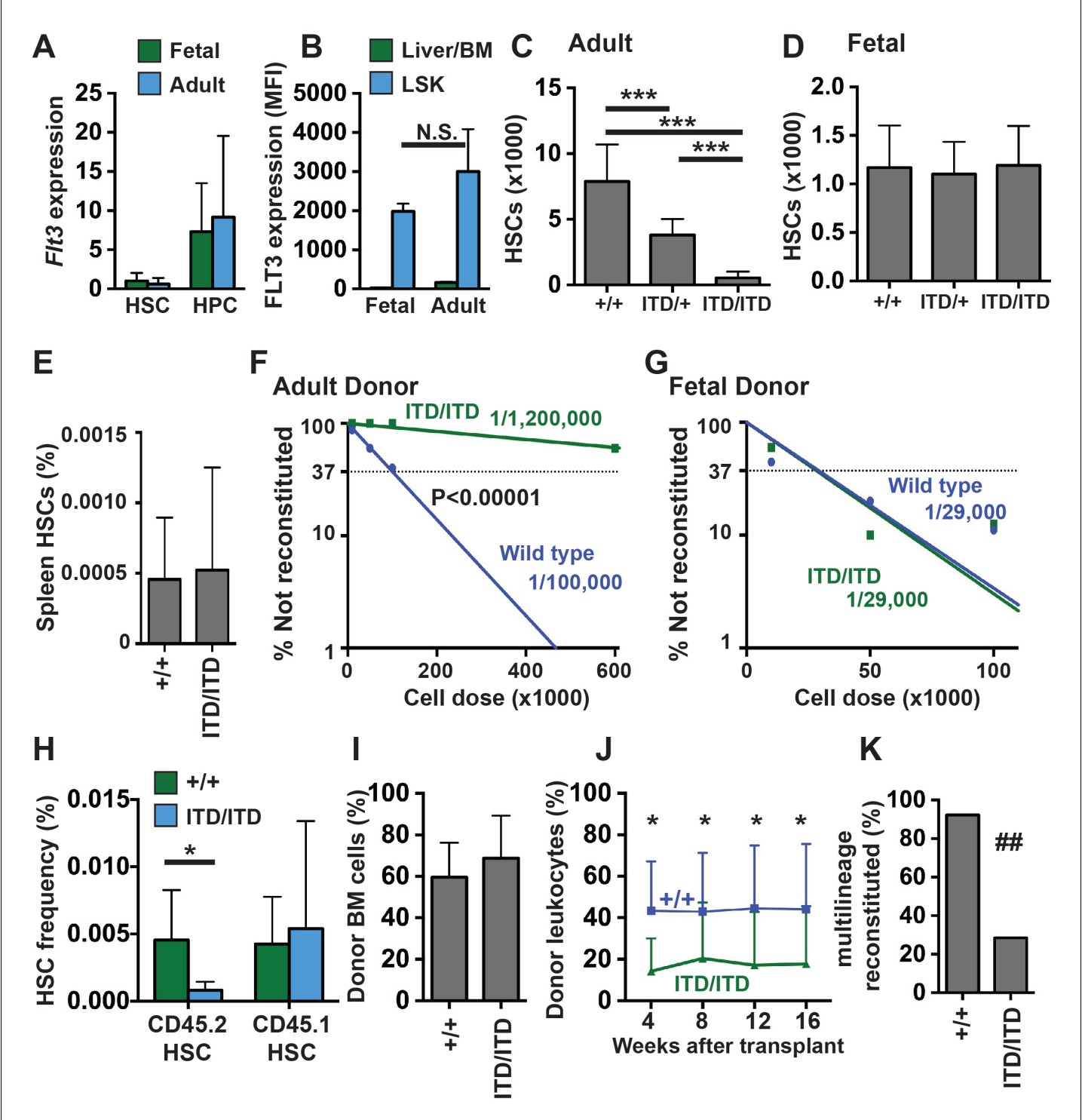

**Figure 1.** *Flt3^ITD* causes HSC depletion in adult but not fetal mice. (A) *Flt3* transcript expression in fetal and adult HSCs and HPCs relative to fetal HSCs; n = 4–9. (B) FLT3 expression in fetal and adult HSC/HPCs (Lineage⁻Sca1⁺c-kit⁺) and unfractionated fetal liver or bone marrow cells, as determined by flow cytometry (N = 3). (C) HSC numbers in two tibias and femurs from adult wild type and *Flt3^ITD* mice; n = 12–16. (D) HSC numbers in fetal livers from E14.5 wild type and *Flt3^ITD* mice; n = 9–20. (E) Spleen HSC frequency in adult wild type and *Flt3^ITD/ITD* mice; n = 4–5. (F,G) Limiting dilution analyses using adult bone marrow (F) or E14.5 fetal liver cells (G); n = 9–10 recipients per cell dose. Wild type and *Flt3^ITD/ITD* HSC frequencies were calculated by extreme limiting dilution analysis. (H,I) Frequencies of donor (CD45.2) and competitor (CD45.1) HSCs (H) and donor bone marrow cells (I) in primary recipients of 100,000 fetal liver cells; n = 15 per genotype. (J) Frequencies of CD45.2+ peripheral blood cells in secondary recipients of donor cells that originated from wild type or *Flt3^ITD/ITD* fetal livers; n = 12–14. (K) Percentage of secondary recipient mice with multilineage donor reconstitution. In all

*Figure 1 continued on next page*

Figure 1 continued

panels, error bars indicate standard deviations and n reflects biological replicates. *p<0.05, ***p<0.001 by two-tailed Student's t-test. ## p<0.01 by Fisher exact probability test.

## $Flt3^{ITD}$ causes HSC depletion and myeloid progenitor expansion by two weeks after birth in mice

We next sought to define the age at which $Flt3^{ITD}$ begins to deplete HSCs and expand myeloid progenitor populations. We measured HSCs, HPCs and granulocyte-monocyte progenitor (GMP) numbers at E14.5, E16.5, post-natal day (P)0 and P14. $Flt3^{ITD/+}$ and $Flt3^{ITD/ITD}$ mice had normal HSC numbers at all ages prior to birth (*Figure 2A*). HSC depletion was evident at P14 in both the bone marrow and the spleen, though not to the extent observed in adult bone marrow (*Figure 2B*). We observed a modest increase in $Flt3^{ITD/+}$ and $Flt3^{ITD/ITD}$ HPCs and GMPs at P0 (*Figure 2C,E*), and this phenotype became more severe, particularly in $Flt3^{ITD/ITD}$ mice, by P14 (*Figure 2D,E*). Spleen enlargement due to MPN was evident by P14 in $Flt3^{ITD/+}$ and $Flt3^{ITD/ITD}$ mice, but E14.5, E16.5 and P0 liver sizes were not increased relative to wild type littermates (*Figure 2F*). These data show that $Flt3^{ITD}$ causes HSC depletion, HPC/GMP expansion and MPN beginning at or shortly after birth.

$FLT3^{ITD}$ mutations usually occur late during the clonal evolution of human AML. This raises the question of whether fetal/neonatal $Flt3^{ITD}$ mice can exhibit HSC depletion and HPC/GMP expansion when a cooperating mutation is present. To test this, we analyzed HSC, HPC and GMP frequencies in $Flt3^{ITD/+}$; $Runx1^{f/+}$; Vav1-Cre mice. Both mono- and bi-allelic RUNX1 loss-of-function mutations co-occur with $FLT3^{ITD}$ in human AML (*Schnittger et al., 2011*), and Runx1 deletions synergize with $Flt3^{ITD}$ to cause AML in mice (*Mead et al., 2013*). For the purposes of these studies, we focused on mono-allelic Runx1 deletions because bi-allelic deletions severely depleted phenotypic HSCs irrespective of the Flt3 genotype (data not shown). These effects were likely due to previously described, Runx1-dependent changes in CD48 expression (*Cai et al., 2011*).

We evaluated HSC, HPC and GMP frequencies in (1) $Flt3^{+/+}$; $Runx1^{f/f}$ or $Runx1^{f/+}$; Cre-negative (control), (2) $Flt3^{ITD/+}$; $Runx1^{f/f}$ or $Runx1^{f/+}$; Cre-negative ($Flt3^{ITD/+}$), (3) $Flt3^{+/+}$;$Runx1^{f/+}$; Vav1-Cre ($Runx1^{\Delta/+}$) and (4) $Flt3^{ITD/+}$;$Runx1^{f/+}$; Vav1-Cre ($Flt3^{ITD/+}$; $Runx1^{\Delta/+}$) littermates at E14.5, P0, P14 and P21. HSCs were severely depleted in P14 and P21 $Flt3^{ITD/+}$; $Runx1^{\Delta/+}$ mice relative to controls and single mutant mice (*Figure 3C,D*). In contrast, all four genotypes of mice had similar HSC frequencies at P0 (*Figure 3B*), and Runx1 heterozygosity increased HSC frequency at E14.5 irrespective of the Flt3 genotype (*Figure 3A*). HPCs and GMPs were markedly expanded in P14 and P21 $Flt3^{ITD/+}$; $Runx1^{\Delta/+}$ mice (*Figure 3G,H,K,L*). These populations were only modestly expanded in compound mutant mice at P0 (*Figure 3F,J*), and they were not expanded at all at E14.5 (*Figure 3E,I*).

We next tested whether compound $Flt3^{ITD}$ and Runx1 mutations had age-specific effects on HSC/HPC function. We transplanted 100,000 P0 liver cells or P21 bone marrow cells from control or $Flt3^{ITD/+}$; $Runx1^{\Delta/+}$ littermate donors, along with 300,000 wild type competitor cells, into irradiated CD45.1 recipient mice. At two weeks after the transplants, we observed CD45.2$^+$ donor-derived leukocytes in the peripheral blood of all recipients, irrespective of donor age or genotype (*Figure 3M, N*). At four weeks after the transplants, donor chimerism was significantly and dramatically reduced in recipients of $Flt3^{ITD/+}$; $Runx1^{\Delta/+}$ P21 donor cells as compared to recipients of control P21 donor cells and $Flt3^{ITD/+}$; $Runx1^{\Delta/+}$ P0 donor cells (*Figure 3M*). Indeed, only 1 of 15 recipients of $Flt3^{ITD/+}$; $Runx1^{\Delta/+}$ P21 donor cells had multi-lineage donor chimerism (>0.5% CD45.2$^+$ myeloid and lymphoid cells) (*Figure 3O*). In contrast, all recipients of control and $Flt3^{ITD/+}$; $Runx1^{\Delta/+}$ P0 donor cells had multi-lineage donor chimerism (*Figure 3O*). These differences were evident even when we focused specifically on myeloid chimerism (*Figure 3N,P*), so they were not simply a reflection of altered lineage biases in the $Flt3^{ITD/+}$; $Runx1^{\Delta/+}$ progenitors. Altogether, these data show that $Flt3^{ITD}$ has developmental context-specific effects on HSC depletion, myeloid progenitor expansion and repopulating activity, even when paired with a cooperating Runx1 mutation.

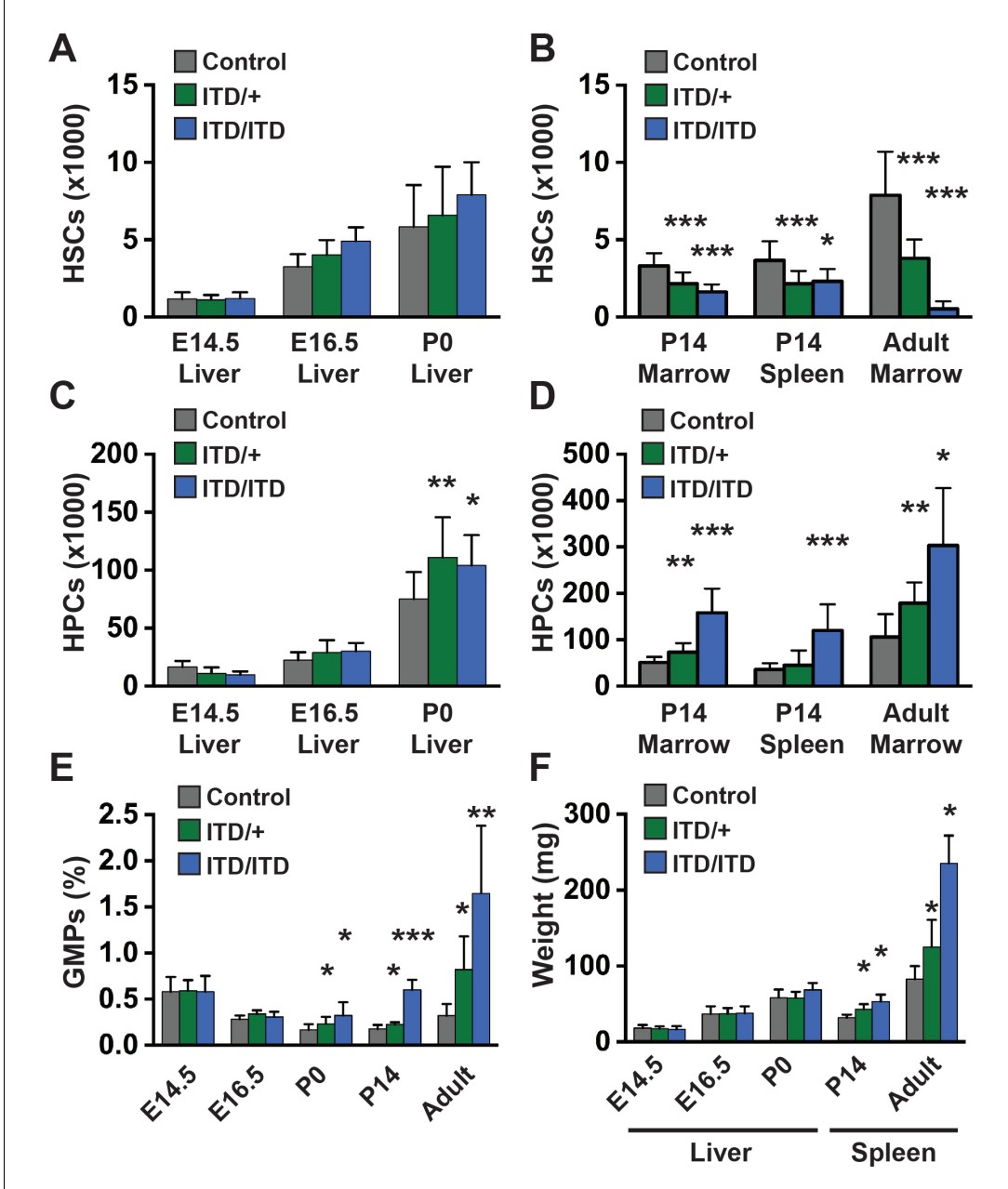

**Figure 2.** *Flt3^{ITD}* causes HSC depletion, HPC expansion and GMP expansion at, or shortly after, birth. (**A**) Absolute HSC numbers in fetal or P0 livers for the indicated genotypes. (**B**) Absolute HSC numbers in P14 and adult bone marrow (two hind limbs) or P14 spleen. (**C,D**) Fetal and adult HPC numbers (two hind limbs). (**E**) GMP frequencies in fetal liver or adult bone marrow. (**F**) Liver or spleen weights. In all panels, error bars indicate standard deviations; n = 6–20 biological replicates for each age and genotype. *p<0.05; **p<0.01; ***p<0.001 by two-tailed Student's t-test relative to the wild type control at the same time point.

### *Flt3^{ITD}* activates STAT5 and MAPK signal transduction pathways in adult HSCs and HPCs, yet it only activates STAT5 in fetal HSCs

To better understand why *Flt3^{ITD}* has developmental context-specific effects on HSCs and HPCs, we sought to better characterize the pathways that mediate *FLT3^{ITD}* signal transduction in vivo. We isolated 25,000 HSC/multipotent progenitors (HSC/MPPs; CD48^-Lineage^-Sca1^+c-kit^+), HPCs and GMPs from adult mice by flow cytometry, and we performed Western blots to assess phosphorylation of

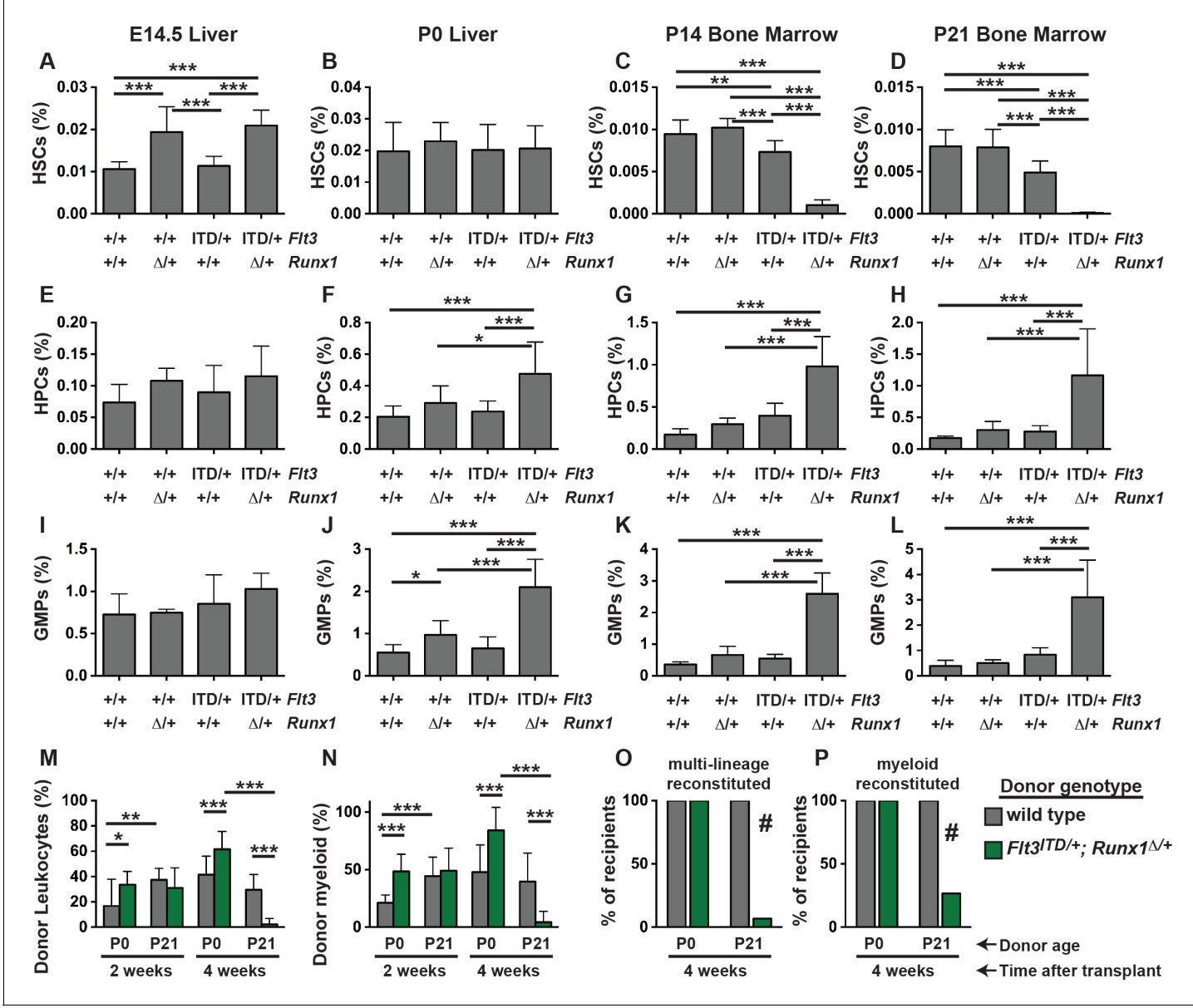

**Figure 3.** *Flt3^ITD* and *Runx1* mutations cooperate to deplete HSCs and expand committed progenitor populations after birth. (A–D) HSC frequencies in E14.5 fetal liver, P0 liver, P14 bone marrow and P21 bone marrow for the indicated genotypes. (E–H) HPC frequencies in E14.5 fetal liver, P0 liver, P14 bone marrow and P21 bone marrow for the indicated genotypes. (I–L) GMP frequencies in E14.5 fetal liver, P0 liver, P14 bone marrow and P21 bone marrow for the indicated genotypes. (M,N) Percentages of CD45.2^+ donor leukocytes (M) or CD11b^+Gr1^+ myeloid cells (N) in the peripheral blood of recipients of P0 liver or P21 bone marrow cells from control or *Flt3^ITD/+; Runx1^Δ/+* mice. Measurements are shown at 2 and 4 weeks after transplantation. (O,P) Percentage of recipients with multi-lineage (O) or myeloid (P) donor reconstitution at four weeks after transplantation. In all panels, error bars indicate standard deviations. For A-L, n = 8–18 biological replicates per genotype and age. For (M–P), n = 14–15 recipients from three independent donors. Statistical significance was determined with a one-way ANOVA followed by Holm-Sidak's post-hoc test for multiple comparisons (*p<0.05; **p<0.01; ***p<0.001), or # p<0.0001 by the Fisher exact probability test.

STAT5, STAT3, ERK1/2 (a MAPK pathway protein) and AKT (a PI3K pathway protein). Both STAT5 and ERK1/2 were hyper-phosphorylated in *Flt3^ITD* HSC/MPPs and HPCs, as well as in adult *Flt3^ITD; Runx1^Δ/Δ* AML cells (*Figure 4A–C*). In contrast, STAT3 and AKT were not hyper-phosphorylated in *Flt3^ITD* mutant HSC/MPPs or HPCs (*Figure 4A* and *Figure 4—figure supplement 1A*), and *Rictor* deletion (PI3K/mTORC2 pathway inactivation) did not rescue *Flt3^ITD*-driven HSC depletion or MPN (*Figure 4—figure supplement 1B,C*). These findings suggest that the STAT5 and MAPK pathways

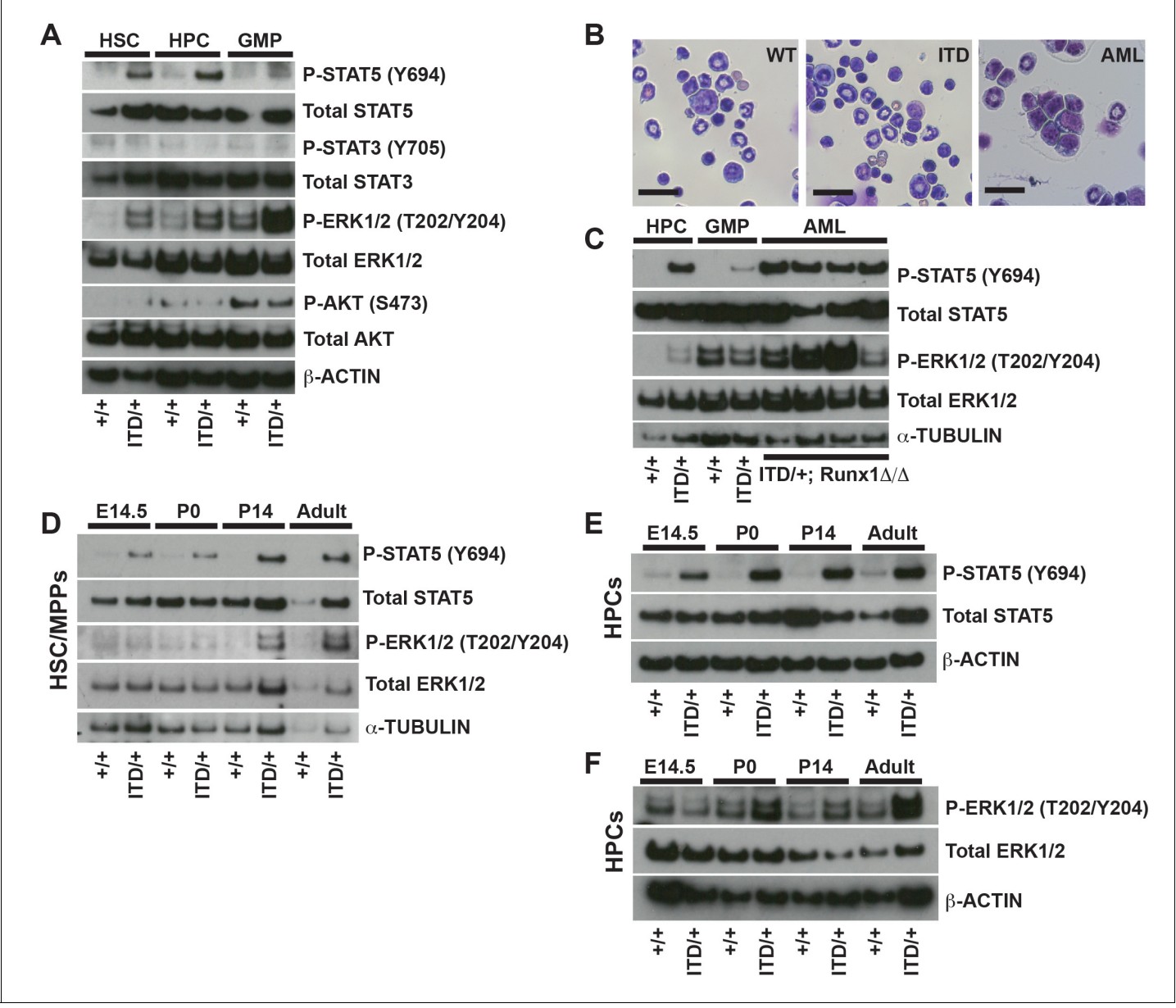

**Figure 4.** FLT3[ITD] activates STAT5 in both fetal and adult progenitors, but it activates the MAPK pathway after birth. (**A**) Western blot showing phosphorylation of STAT5, ERK1/2, STAT3 and AKT in adult wild type and *Flt3[ITD/+]* HSC/MPPs, HPCs and GMPs. (**B**) *Flt3[ITD/+]; Runx1[Δ/Δ]* progenitors give rise to AML in adult mice (right panel) that is not observed in wild type or *Flt3[ITD/+]* bone marrow. Scale bars indicate 100 microns. (**C**) STAT5 and MAPK are hyper-phosphorylated in *Flt3[ITD/+]; Runx1[Δ/Δ]* AML that develops in adult mice. (**D**) STAT5 and ERK1/2 phosphorylation in wild type and *Flt3[ITD/+]* HSC/MPPs at E14.5, P0, P14 and adulthood. (**E**) STAT5 phosphorylation in wild type and *Flt3[ITD/+]* HPCs at E14.5, P0, P14 and adulthood. (**F**) ERK1/2 phosphorylation in wild type and *Flt3[ITD/+]* HPCs at E14.5, P0, P14 and adulthood. Each blot is representative of two (panels **A** and **C**) or at least three (panels **D–F**) independent experiments.

The following figure supplement is available for figure 4:

**Figure supplement 1.** The PI3K/mTORC2 pathway does not mediate HSC depletion or MPN in *Flt3[ITD]* mutant mice.

mediate FLT3[ITD] signal transduction in hematopoietic progenitors, but the STAT3 and PI3K pathways do not.

We next tested whether FLT3[ITD] signal transduction changes between fetal and adult stages of development. We isolated HSC/MPPs and HPCs from wild type and *Flt3[ITD]* mice at E14.5, P0, P14

and eight weeks after birth. We performed Western blots to assess STAT5 and ERK1/2 phosphorylation. STAT5 was hyper-phosphorylated in $Flt3^{ITD}$ mutant HSC/MPPs and HPCs at all stages of development, though the degree of STAT5 phosphorylation appeared to increase with age (*Figure 4D,E*). ERK1/2 was only hyper-phosphorylated in post-natal $Flt3^{ITD}$ mutant HSC/MPPs and HPCs (*Figure 4D,F*). Several other signal transduction proteins, including STAT3, AKT, ribosomal protein S6, p38 and JNK, were not hyper-phosphorylated in $Flt3^{ITD}$ HSC/MPPs or HPCs at any age tested, or their phosphorylation was undetectable (data not shown). Our data reinforce other studies that have implicated STAT5 and MAPK as key downstream effectors of FLT3$^{ITD}$ signaling (*Choudhary et al., 2007*; *Radomska et al., 2006*). However, the data suggest that these pathways are not coupled — STAT5 is phosphorylated in fetal progenitors without concurrent MAPK pathway activation. This raises the question of whether each pathway has unique functions downstream of FLT3$^{ITD}$.

## MAPK pathway inhibition has little to no effect on $Flt3^{ITD}$-driven HSC depletion, HPC expansion and GMP expansion

We used the MEK inhibitor PD0325901 to test whether MAPK pathway inhibition could prevent HSC depletion and HPC/GMP expansion in $Flt3^{ITD}$ mice. We administered vehicle or PD0325901 to 6-week-old wild type and $Flt3^{ITD/+}$ mice (5 mg/kg per day for 10 days). This regimen effectively inhibited ERK1/2 phosphorylation in HPCs without affecting STAT5 phosphorylation (*Figure 5—figure supplement 1*). PD0325901-treated wild type mice had significantly more phenotypic HSCs and HPCs than vehicle treated controls (*Figure 5A,B*). However, PD0325901 had no effect on HSC numbers, HPC numbers or GMP frequencies in $Flt3^{ITD/+}$ mice (*Figure 5A–C*). This suggests that sustained MAPK pathway signaling is not required for HSC depletion, HPC expansion and GMP expansion in $Flt3^{ITD/+}$ adult mice.

We next tested whether PD0325901 could prevent the onset of the HSC depletion, HPC expansion and GMP expansion phenotypes if it was given shortly after birth. We treated nursing mothers of wild type, $Flt3^{ITD/+}$ and $Flt3^{ITD/ITD}$ neonates with PD0325901 (5 mg/kg per day) beginning at P1. While this regimen has previously been shown to rescue MAPK pathway-dependent developmental abnormalities in $Nf1$ mutant neonates (*Wang et al., 2012*), it did not prevent HSC depletion or HPC expansion in $Flt3^{ITD}$ mutant neonates (*Figure 5D,E*), and it only partially rescued GMP expansion (*Figure 5F*). Altogether, the data suggest that the MAPK pathway has only a minor role, if any, in causing these phenotypes. Temporal changes in MAPK pathway regulation are unlikely to account for the different effects of FLT3$^{ITD}$ on fetal and adult progenitors.

## STAT5 inactivation exacerbates HSC depletion, HPC expansion, GMP expansion and MPN

STAT5 has been implicated as a key downstream effector of FLT3$^{ITD}$ in many different systems, and it is hyper-phosphorylated in $Flt3^{ITD/+}$ HSCs and HPCs during fetal, neonatal and adult stages of development (*Figure 4E,F*). This raised the question of whether genetic inactivation of $Stat5a$ and $Stat5b$ – with a conditional $Stat5a/b$ allele (*Wang et al., 2009*)— could prevent HSC depletion, HPC expansion, GMP expansion and MPN in $Flt3^{ITD/+}$ mice. To answer this question, we evaluated HSCs, HPCs, GMPs and spleen weights in 1) $Flt3^{+/+}$; $Stat5a/b^{f/+}$ or $Stat5a/b^{f/f}$; Cre$^-$ (control), 2) $Flt3^{+/+}$; $Stat5a/b^{f/+}$; $Mx1$-Cre ($Stat5^{\Delta/+}$), 3) $Flt3^{+/+}$; $Stat5a/b^{f/f}$; $Mx1$-Cre ($Stat5^{\Delta/\Delta}$), 4) $Flt3^{ITD/+}$; $Stat5a/b^{f/+}$ or $Stat5a/b^{f/f}$; Cre$^-$ ($Flt3^{ITD/+}$), 5) $Flt3^{ITD/+}$; $Stat5a/b^{f/f}$; $Mx1$-Cre ($Flt3^{ITD}$; $Stat5^{\Delta/+}$), and 6) $Flt3^{ITD/+}$; $Stat5a/b^{f/f}$; $Mx1$-Cre ($Flt3^{ITD/+}$; $Stat5^{\Delta/\Delta}$) mice. The mice were treated with poly-inosine:poly-cytosine (pIpC) beginning at six weeks after birth to delete $Stat5a/b$, and they were analyzed four weeks later. Western blotting confirmed complete loss of STAT5 protein, and MAPK pathway activation was unaffected by $Stat5a/b$ deletion (*Figure 6—figure supplement 1*). Surprisingly, $Stat5a/b$ deletion exacerbated the HSC depletion, HPC expansion and GMP expansion phenotypes of $Flt3^{ITD/+}$-mice rather than rescuing them (*Figure 6A–C*). Spleen weights were also enlarged in $Flt3^{ITD}$; $Stat5^{\Delta/+}$ and $Flt3^{ITD/+}$; $Stat5^{\Delta/\Delta}$ mice relative to control, $Stat5^{\Delta/+}$, $Stat5^{\Delta/\Delta}$ and $Flt3^{ITD/+}$ littermates (*Figure 5D*). Similar results were observed when we deleted a single $Stat5a/b$ with $Vav1$-cre. Only one $Stat5a/b$ allele was deleted in these analyses because bi-allelic deletion impairs fetal erythropoiesis (*Zhu et al., 2008*). Nevertheless, $Flt3^{ITD/+}$; $Stat5a/b^{f/+}$; $Vav1$-Cre mice had fewer HSCs, more

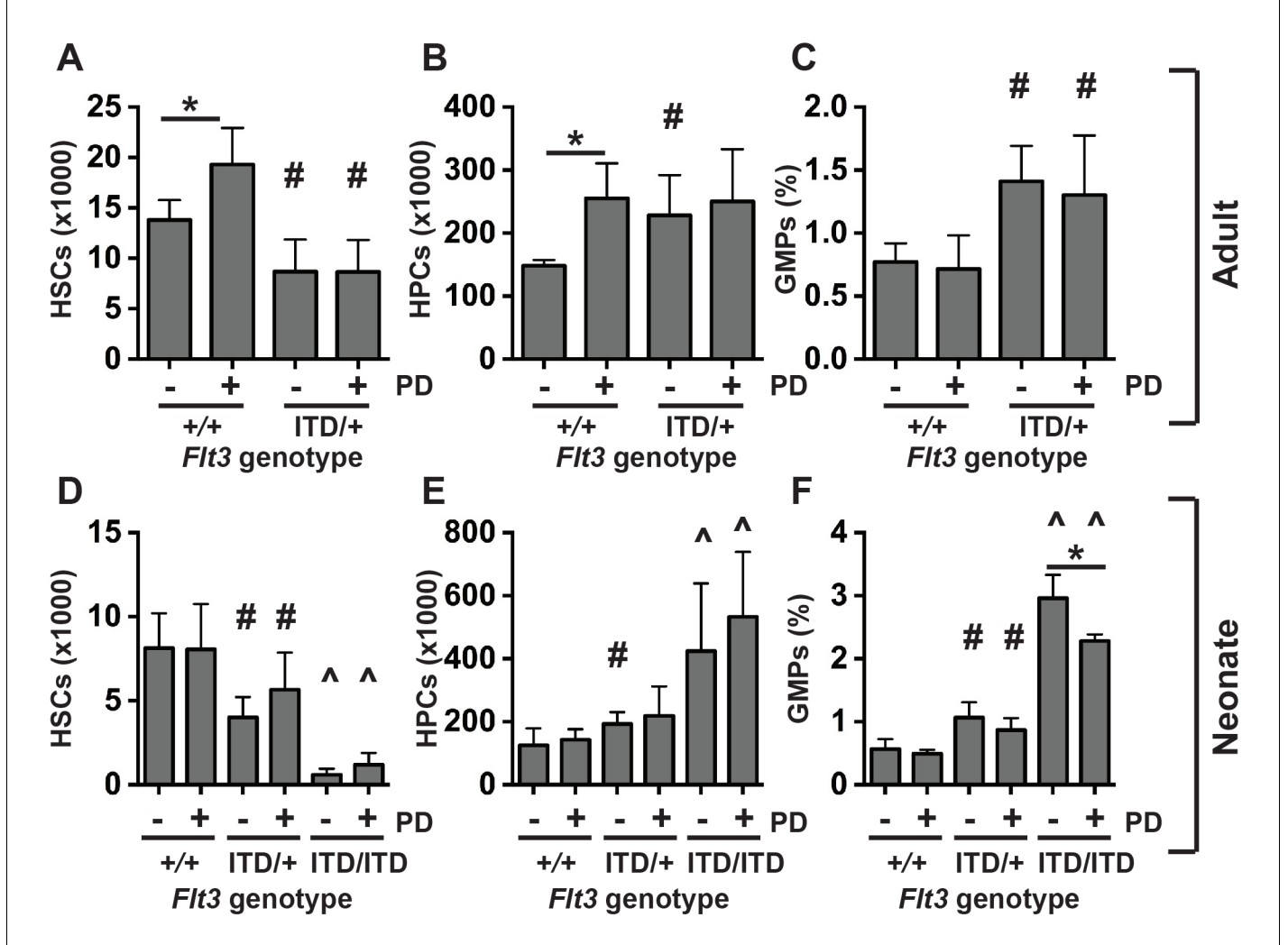

**Figure 5.** MAPK pathway inhibition does not prevent HSC depletion or committed progenitor expansion in *Flt3*$^{ITD/+}$ mice. (A–D) HSC numbers (A), HPC numbers (B) and GMP frequencies (C) in wild type and *Flt3*$^{ITD/+}$ mice that were treated with vehicle or PD0325901 for 10 days beginning at six weeks after birth; n = 4–5 biological replicates per genotype and treatment. (D–F) HSC numbers (D), HPC numbers (E) and GMP frequencies (F) in P19 wild type, *Flt3*$^{ITD/+}$ and *Flt3*$^{ITD/ITD}$ mice whose mothers were given PD0325901 beginning at P1; n = 4–15 biological replicates for each genotype and treatment. In all panels, error bars indicate standard deviation. Statistical comparisons were made with a two-tailed Student's t-test. *p<0.05 relative to vehicle treated cells with equivalent genotypes; # p<0.05 relative to similarly treated wild type controls; ˄p<0.05 relative to similarly treated wild type and *Flt3*$^{ITD/+}$ groups.

The following figure supplement is available for figure 5:

**Figure supplement 1.** Inhibition of the MAPK pathway fails to rescue FLT3$^{ITD}$-mediated HSC depletion and myeloid progenitor expansion, but *Stat5a/b* deletion enhances these phenotypes.

HPCs, more GMPs and larger spleens than control or *Flt3*$^{ITD/+}$ littermates at 8–10 weeks after birth (*Figure 6E–H*).

The data reveal an unanticipated function for STAT5 in pre-leukemic, *Flt3*$^{ITD}$-mutant progenitors. They suggest that STAT5 helps to maintain *Flt3*$^{ITD}$-mutant HSCs in an uncommitted state and that it antagonizes *Flt3*$^{ITD}$-driven expansion of more committed myeloid progenitor populations. Thus, FLT3$^{ITD}$ may simultaneously potentiate self-renewal and myeloid commitment programs via STAT5-dependent and STAT5-independent pathways, respectively (*Figure 6I*).

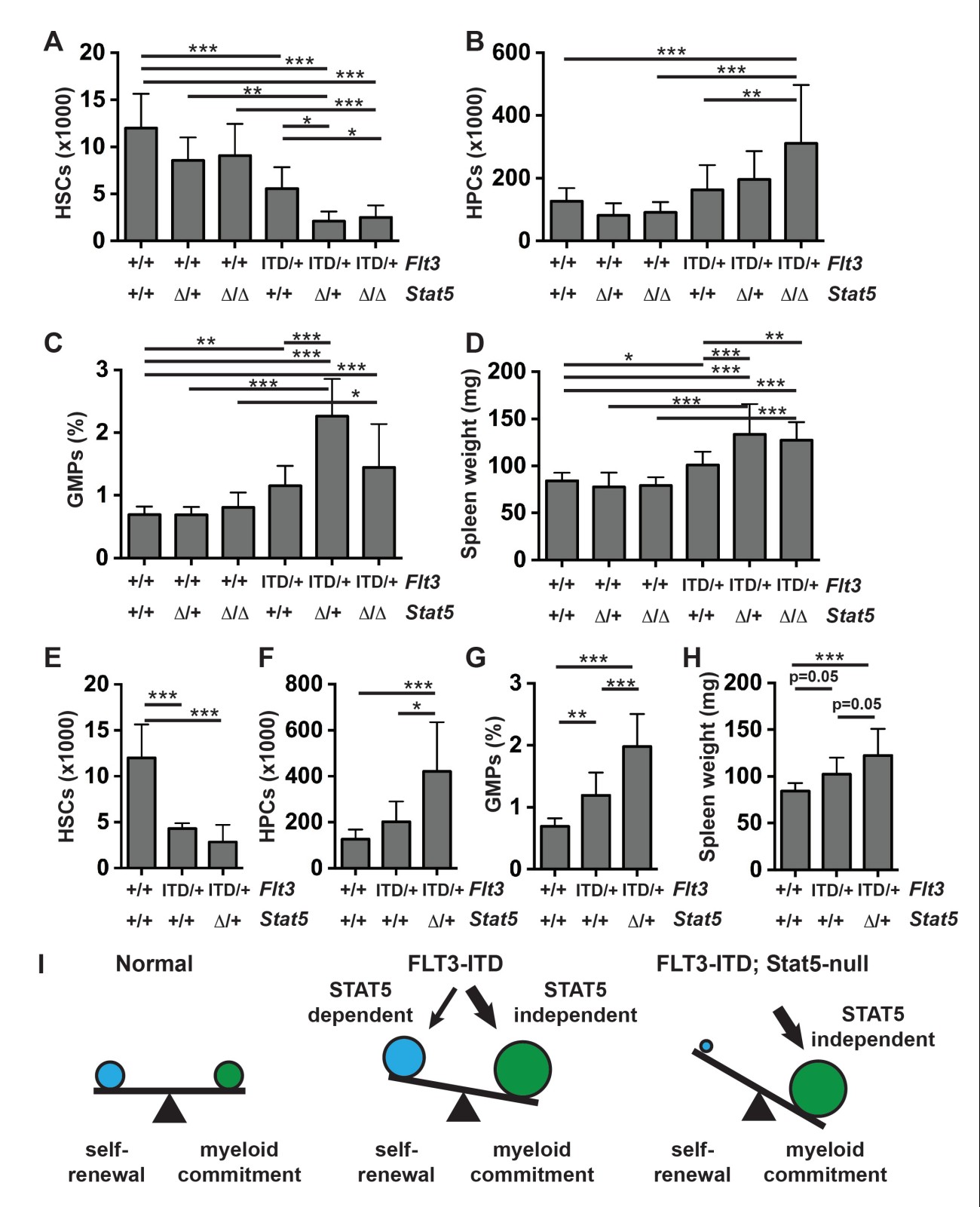

**Figure 6.** *Stat5a/b* deletion exacerbates rather than rescues HSC depletion, HPC expansion, GMP expansion and MPN in *Flt3^ITD/+* mice. (A–D) HSC numbers (A), HPC numbers (B), GMP frequencies (C) and spleen weights (D) in *Flt3^ITD/+; Stat5a/b^f/f; Mx1-Cre* compound mutant mice and littermate controls; n = 6–20 biological replicates per genotype. *Stat5a/b* was conditionally deleted six weeks after birth, and this caused a complete loss of protein expression (figure supplement 1). (E–H) HSC numbers (E), HPC numbers (F), GMP frequencies (G) and spleen weights (H) in *Flt3^ITD/+; Stat5a/*

*Figure 6 continued on next page*

Figure 6 continued

$b^{f/+}$; Vav1-Cre compound mutant mice and littermate controls; n = 8–20 biological replicates per genotype. (I) The data suggest that STAT5-dependent pathways promote HSC self-renewal downstream of FLT3$^{ITD}$, but these effects are outweighed by STAT5-independent myeloid commitment pathways. In all panels, error bars indicate standard deviation. Statistical significance was determined with a one-way ANOVA followed by Holm-Sidak's post-hoc test for multiple comparisons. *p<0.05; **p<0.01; ***p<0.001.

The following figure supplement is available for figure 6:

**Figure supplement 1.** Stat5a/b deletion causes a complete loss of phosphorylated and total STAT5 protein.

## FLT3$^{ITD}$ activates self-renewal programs in HPCs via STAT5, and it activates commitment programs independently of STAT5

The changes in HSC and HPC frequencies in Flt3$^{ITD}$; Stat5a/b compound mutant mice raise the question of whether FLT3$^{ITD}$ has STAT5-dependent and STAT5-independent effects on gene expression, and whether transcriptional changes are developmental context-specific. To answer these questions we performed two independent experiments to characterize global changes in gene expression (*Figure 7A*). In the first experiment, we analyzed gene expression in wild type and Flt3$^{ITD/+}$ HSCs and HPCs at E14.5, P0, P14 and 8–10 weeks after birth. This experiment was meant to elucidate changes in FLT3$^{ITD}$ target genes over time. In the second experiment, we analyzed gene expression in adult HPCs from (1) wild type, (2) Flt3$^{ITD/+}$, (3) Flt3$^{ITD/ITD}$, (4) Flt3$^{ITD}$; Stat5$^{Δ/+}$ and 5) Flt3$^{ITD/+}$; Stat5$^{Δ/Δ}$ mice. This experiment was meant to delineate which FLT3$^{ITD}$ targets are STAT5-dependent and which are STAT5-independent.

We analyzed the data from each experiment independently, and we merged the data to identify a list of genes that were significantly, differentially expressed in FLT3$^{ITD}$ progenitors in both experiments. In experiment 1, we identified 254 annotated coding genes that were differentially expressed between wild type and Flt3$^{ITD/+}$ HPCs at one or more time points (*Figure 7—source data 1*; adjusted p<0.05; fold change ≥ 2). We did not identify any genes that met these stringent filtering criteria in HSCs, though statistically significant changes in gene expression were observed when specific target genes (from the HPC list) were individually interrogated (p<0.05, *Figure 7—source data 1*). The differences between HSCs and HPCs may simply reflect differences in Flt3 expression (*Figure 1A*), though it is also possible that HSCs with the strongest transcriptional responses to FLT3$^{ITD}$ were not captured in our microarray assays because they differentiated. Of the 254 genes that were differentially expressed in experiment 1, 58 unique genes were also differentially expressed between wild type and Flt3$^{ITD/+}$ HPCs in experiment 2 (*Figure 7B* and *Figure 7—figure supplement 1*). Thirty-three genes were expressed at higher levels in Flt3$^{ITD}$ HPCs relative to wild type HPCs, and 25 genes were expressed at lower levels (*Figure 7B*). Of these, 35 normalized when Stat5a/b was deleted, but 23 did not. FLT3$^{ITD}$ therefore has both STAT5-dependent and STAT5-independent effects on gene expression, and these effects are more pronounced in HPCs as compared to HSCs.

We tested whether FLT3$^{ITD}$ activates self-renewal- and commitment-related transcriptional programs via STAT5-dependent and STAT5-independent mechanisms, respectively, as predicted by our phenotypic assays (*Figure 6*). We generated self-renewal-related and commitment-related gene sets by comparing wild type HSCs and HPCs using the data collected in experiment 1 (*Figure 7—source data 2*). We then used Gene Set Enrichment Analysis (GSEA) to compare wild type, Flt3$^{ITD/+}$ and Flt3$^{ITD/+}$; Stat5$^{Δ/Δ}$ HPCs using data collected in experiment 2 (*Subramanian et al., 2005*). Self-renewal-related genes were enriched in Flt3$^{ITD/+}$ HPCs, and commitment-related genes were enriched in wild type HPCs (*Figure 7C*). This suggests that the FLT3$^{ITD}$ protein can activate self-renewal mechanisms in otherwise non-self-renewing HPCs. Remarkably, these effects were strongly reversed when Stat5a/b was deleted (*Figure 7C*). A separately curated self-renewal gene set from Ivanova et al. was similarly enriched in wild type and Flt3$^{ITD/+}$ HPCs as compared to Flt3$^{ITD/+}$; Stat5$^{Δ/Δ}$ HPCs (*Figure 7D*) (*Ivanova et al., 2002*). These findings are consistent with a model in which FLT3$^{ITD}$ signals via STAT5 to ectopically activate self-renewal programs in HPCs, and it simultaneously promotes myeloid commitment via STAT5-independent mechanisms.

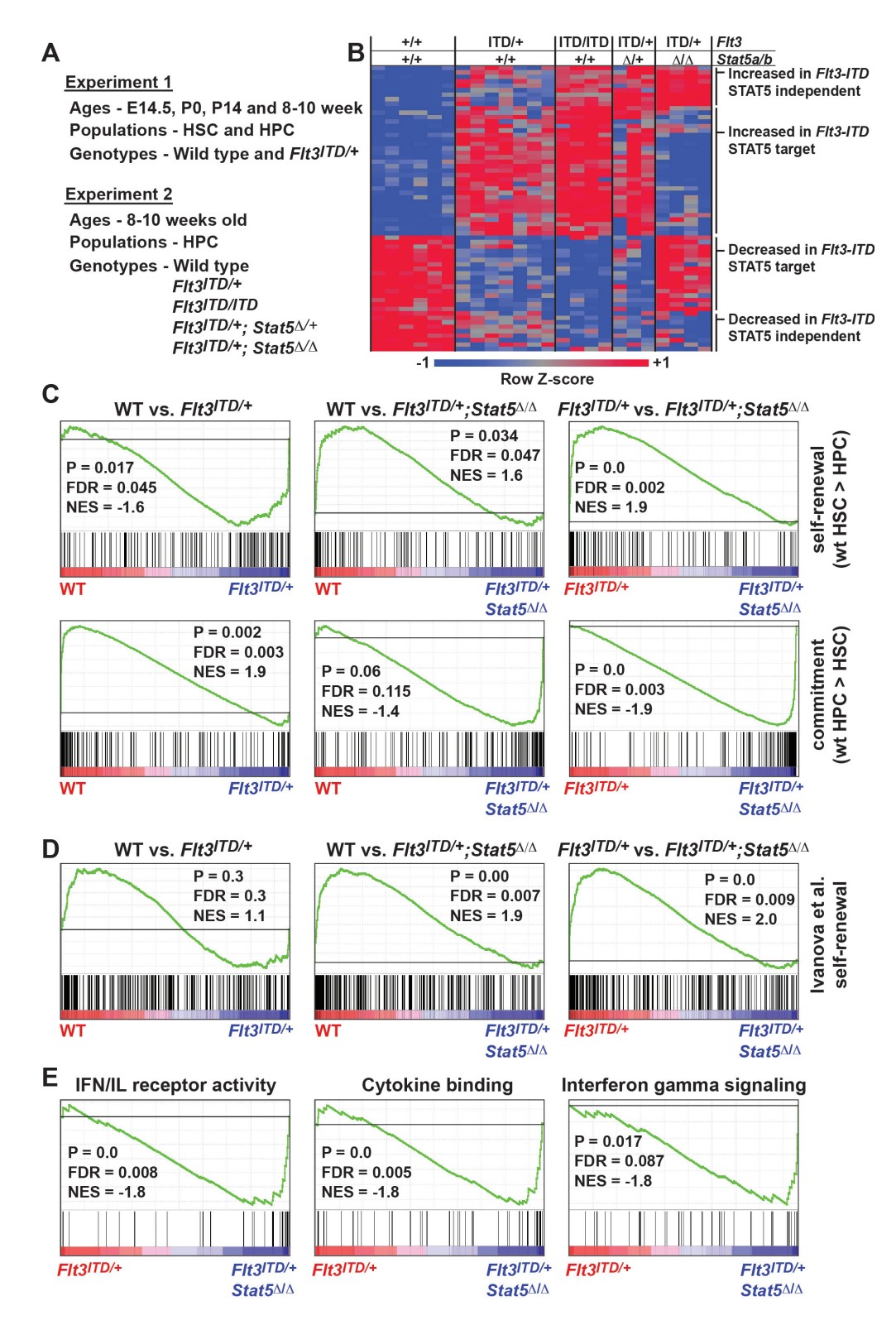

**Figure 7.** FLT3[ITD] activates STAT5-dependent self-renewal programs and STAT5-independent commitment programs. (**A**) Overview of experimental design. (**B**) Heatmap representing genes that were differentially expressed in *Flt3[ITD]* mutant HPCs relative to wild type HPCs in both experiments 1 and 2. Each column represents an independent sample. The gene names and dendrogram are shown in *Figure 7—figure supplement 1* attached to this figure. (**C**) Self-renewal and commitment-related gene sets were generated by identifying genes that were more highly expressed (>5 fold, adj. p<0.05)

*Figure 7 continued on next page*

*Figure 7 continued*

in HSCs relative to HPCs (self-renewal), or HPCs relative to HSCs (commitment). GSEA plots show ectopic activation of self-renewal-related genes in HPCs that express FLT3^ITD, but these effects are reversed in *Stat5a/b*-deficient HPCs. (D) An independently curated self-renewal gene set (*Ivanova et al., 2002*) was similarly enriched in wild type and *Flt3^ITD/+* HPCs relative to *Flt3^ITD/+; Stat5^Δ/Δ* HPCs. (E) GSEA revealed enrichment of gene sets associated with increased inflammatory cytokine signaling.

The following source data and figure supplement are available for figure 7:

**Source data 1.** Significantly differentially expressed genes in *Flt3^ITD/+* HSCs and HPCs.
**Source data 2.** Self-renewal-related and commitment-related gene sets.
**Figure supplement 1.** FLT3^ITD induces STAT5-dependent and STAT5-independent changes in gene expression.

To better understand the STAT5-independent mechanisms that promote myeloid commitment, we performed GSEA on *Flt3^ITD/+* and *Flt3^ITD/+; Stat5^Δ/Δ* HPCs with curated gene sets in the MSigDB database (*Subramanian et al., 2005*). The most significantly enriched gene sets in *Flt3^ITD/+; Stat5^Δ/Δ* HPCs were generally associated with increased inflammatory cytokine signaling (*Figure 7E*). This finding is intriguing because several prior studies have linked inflammatory cytokine signaling to loss of adult HSC self-renewal capacity and myeloid differentiation (*Baldridge et al., 2010*; *Essers et al., 2009*; *Pietras et al., 2016*). Of note, we did not observe changes in STAT1, STAT3 or AKT phosphorylation in *Flt3^ITD/+; Stat5^Δ/Δ* HPCs (*Figure 6—figure supplement 1* and data not shown). Additional studies are still needed to identify the signal transduction molecules that mediate FLT3^ITD-driven myeloid commitment, and to test whether changes in cytokine-related gene expression are a cause or a consequence of differentiation in *Flt3^ITD/+; Stat5^Δ/Δ* HPCs.

## FLT3^ITD-mediated changes in gene expression correlate temporally with the normal transition from fetal to adult transcriptional programs

FLT3^ITD has the capacity to activate functionally relevant signal transduction pathways, such as STAT5, during both pre- and post-natal stages of development (*Figure 4*), yet HSC and HPC phenotypes were only observed after birth. This raises the question of whether pre- and post-natal progenitors have distinct transcriptional responses to FLT3^ITD. We evaluated FLT3^ITD target gene expression in HPCs at E14.5, P0, P14 and adulthood (*Figure 8A*). We found that differences between wild type and *Flt3^ITD/+* HPCs were more evident at P14 and adulthood than at E14.5 or P0 (*Figure 8A* and *Figure 8—figure supplement 1*). This was true for both STAT5-dependent targets, e.g. *Socs2*, and STAT5-independent targets, e.g. *Ctsg* (*Figure 8B*). Thus, fetal, neonatal and adult hematopoietic progenitors have distinct transcriptional responses to FLT3^ITD signaling.

To better understand when HSCs and HPCs transition from fetal to adult transcriptional programs, we analyzed gene expression in wild type cells from experiment 1. We identified 2627 unique genes (from 3005 different probes) that exhibited significant changes in gene expression in HSCs between E14.5 and adulthood. Of the 228 most differentially expressed transcripts (from the top 250 probes), all followed a consistent trend toward increasing (109 genes) or decreasing (119 genes) expression with increasing age, and most were differentially expressed in both HSCs and HPCs (*Figure 8C*, *Figure 8—source data 1*). Among these genes were several that encode transcription factors and RNA binding proteins that are known to regulate HSC self-renewal, including *Lin28b*, *Esr1, Hmga2* and *Egr1* (*Figure 8C,D*). Principal component analysis and Euclidean distance measurements showed that P14 HSCs more closely resembled adult HSCs than fetal HSCs, and P0 HSCs more closely resembled fetal HSCs (*Figure 8E*). Similar associations were observed for HPCs (*Figure 8—figure supplement 2*). The data show that HSCs and HPCs begin transitioning from fetal to adult transcriptional programs by P14, even before they achieve quiescence (*Bowie et al., 2006*). Furthermore, the data suggest that HSCs and HPCs become competent to express (or repress) FLT3^ITD target genes as they transition from fetal to adult transcriptional programs.

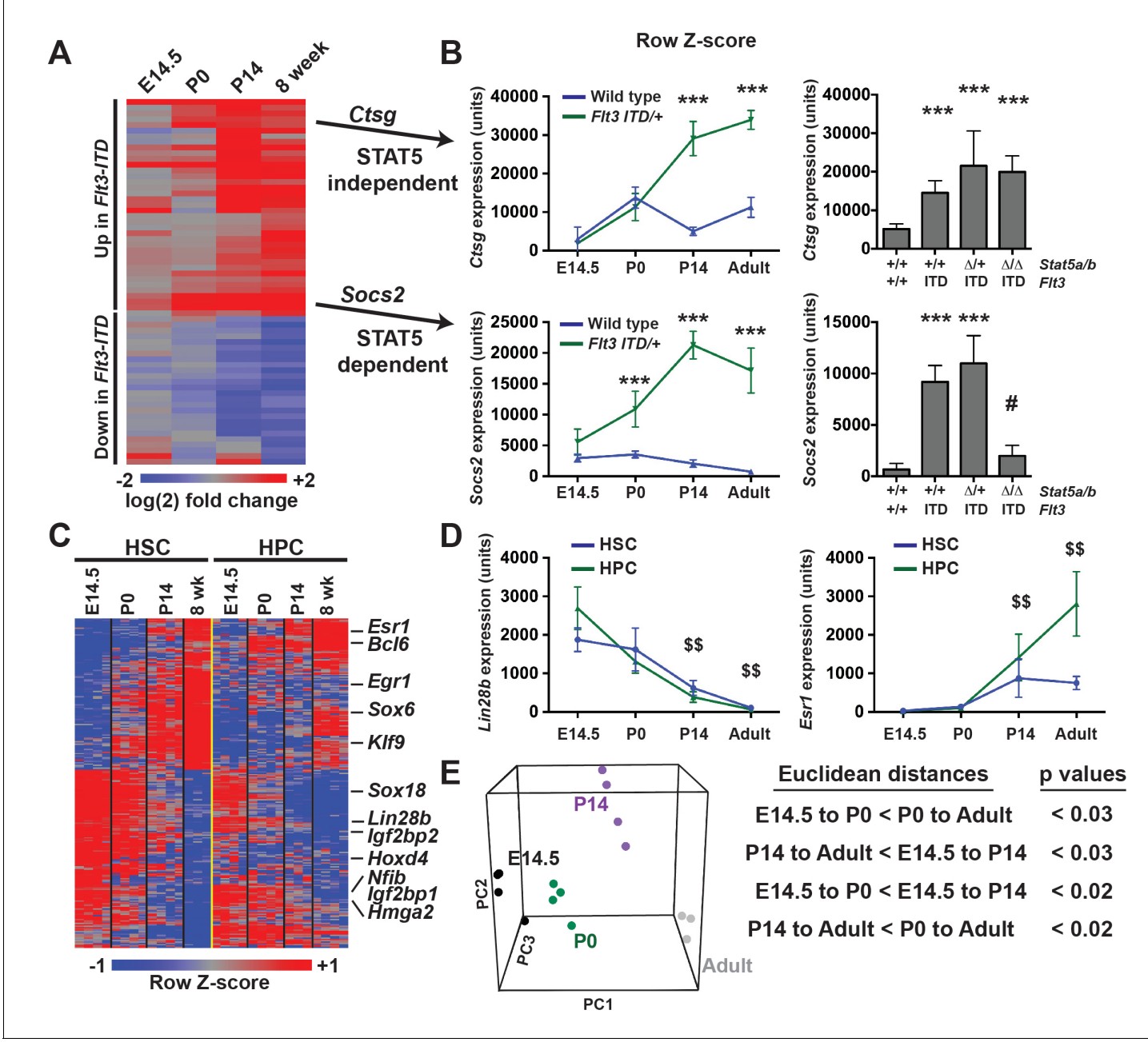

**Figure 8.** *Flt3^ITD*-mediated changes in gene expression correlate with the normal transition from fetal to adult transcriptional programs. (**A**) Heatmap showing expression of FLT3^ITD target genes at E14.5, P0, P14 and adult stages. Each column shows average fold change in *Flt3^ITD/+* HPCs relative wild type HPCs at the indicated time point; n = 3–4 independent arrays per genotype. The gene names and dendrogram are shown in figure supplement one attached to this figure. (**B**) Representative examples of expression of STAT5-independent (*Ctsg*) and STAT5-dependent (*Socs2*) FLT3^ITD targets. Error bars reflect standard deviation. ***adj. p<0.05 relative to wild type at the same time point, # adj. p<0.05 relative to *Flt3^ITD/+* at the same time point. (**C**) Heterochronic genes began transitioning from fetal to adult expression patterns between P0 and P14, concordant with sensitivity to FLT3^ITD. Genes that encode transcription factors and RNA binding proteins are noted to the right of the heatmap. A complete gene list is provided in *Figure 8—source data 1* attached to this figure. (**D**) Representative examples of heterochronic genes that show decreased (*Lin28b*) or increased (*Esr1*; Estrogen Receptor α) expression in adult relative to fetal HSCs and HPCs. Error bars reflect standard deviation. $$ adj. p<0.05 relative to E14.5 for both HSCs and HPCs. (**E**) Principal component analysis and Euclidean distance measurements show that gene expression in P0 HSCs more closely resembles fetal HSCs than adult HSCs, and gene expression in P14 HSCs more closely resembles that of adult HSCs. Similar calculations for HPCs are shown in figure supplement 2.

The following source data and figure supplements are available for figure 8:

*Figure 8 continued on next page*

*Figure 8 continued*

**Source data 1.** Heterochronic gene expression in wild type HSCs and HPCs.
**Figure supplement 1.** FLT3^ITD-mediated changes in gene expression correlate temporally with a transition from fetal to adult transcriptional states.
**Figure supplement 2.** HPCs express heterochronic genes and begin to transition from fetal to adult transcriptional programs by P14.

## *Flt3^ITD* and *Runx1* heterozygous mutations collaboratively induce changes in gene expression in a developmental context-dependent manner

*Flt3^ITD* and *Tet2* loss of function mutations have recently been shown to cooperatively induce changes in gene expression and DNA methylation in adult HPCs that are not observed with either mutation alone (***Shih et al., 2015***). We tested whether *Flt3^ITD* and *Runx1* mutations have similar cooperative effects on transcription and whether the effects are age-specific. We evaluated gene expression in (1) wild type, (2) *Runx1^Δ/+*, (3) *Flt3^ITD/+* and (4) *Flt3^ITD/+*; *Runx1^Δ/+* HPCs at P0 and P14. At P14, we identified 191 genes that were significantly differentially expressed between wild type and *Flt3^ITD/+*; *Runx1^Δ/+* HPCs (adj. p<0.05; fold change ≥ 3). At P0 only eight genes met these criteria, seven of which overlapped with the P14 gene list (***Figure 9A*** and ***Figure 9—source data 1***). GSEA showed significant overlap between genes that were differentially expressed in *Flt3^ITD/+*; *Runx1^Δ/+* HPCs and those that Shih et al. found to be differentially expressed in *Flt3^ITD*; *Tet2^Δ/Δ* HPCs (***Shih et al., 2015***) (***Figure 9B***). This suggests that FLT3^ITD can cooperate with diverse mutations to induce a conserved set of target genes.

We used hierarchical clustering to better visualize differences in gene expression at P0 and P14. This approach did show some differences between wild type and *Flt3^ITD/+*; *Runx1^Δ/+* HPCs at P0 (***Figure 9C***), but much greater differences were observed at P14 for most target genes (***Figure 9C, D***). Of the 25 most differentially expressed genes in P14 *Flt3^ITD/+*; *Runx1^Δ/+* HPCs, only three – *Nov*, *Bhlhe40* and *Mboat2* – were induced equally at both P0 and P14 (***Figure 9D***, ***Figure 9—source data 1***). The remaining genes showed only a partial change in expression at P0 (e.g. *Socs2*) or no change in expression (e.g. *Adgre1*, *Dusp6*, *Gem*, *Postn*) (***Figure 9D***). GSEA also showed differences in the transcriptional programs of P0 and P14 *Flt3^ITD/+*; *Runx1^Δ/+* HPCs. Several Gene Ontology and Oncogenic Signature gene sets were enriched in P14 *Flt3^ITD/+*; *Runx1^Δ/+* HPCs relative to wild type controls, the most significant of which included genes that are negatively regulated by mTOR (***Majumder et al., 2004***), genes that inhibit apoptosis (Gene Ontology) and c-myc targets (***Bild et al., 2006***) (***Figure 9E***). These gene sets were not significantly enriched in P0 HSCs with the exception of the anti-apoptotic gene set, which was paradoxically enriched in wild type HPCs relative to *Flt3^ITD/+*; *Runx1^Δ/+* (***Figure 9E***). Altogether, the data show that cooperating *Flt3^ITD* and *Runx1* mutations — and likely other cooperating mutations — have developmental context-specific effects on gene expression.

## Discussion

Our data offer a potential explanation for why FLT3^ITD mutations are rare in infants and young children with AML. The mutant FLT3^ITD protein hyper-activates STAT5 in hematopoietic progenitors during both fetal and adult stages of development, yet its effects on transcription are realized selectively in adult progenitors. Even cooperative interactions between *Flt3^ITD* and *Runx1* heterozygous mutations are blunted during fetal and perinatal stages. Thus, FLT3^ITD mutations may occur disproportionately in older children and adults with AML because they are less able to activate key effectors of leukemogenesis during earlier stages of life.

## *Flt3^ITD* has distinct STAT5-dependent and STAT5-independent effects on self-renewal and myeloid commitment, respectively

In the course of these studies, we discovered that *Stat5a/b* deletion exacerbates, rather than rescues, the HSC depletion, HPC/GMP expansion and MPN phenotypes of *Flt3^ITD/+* mice. Furthermore,

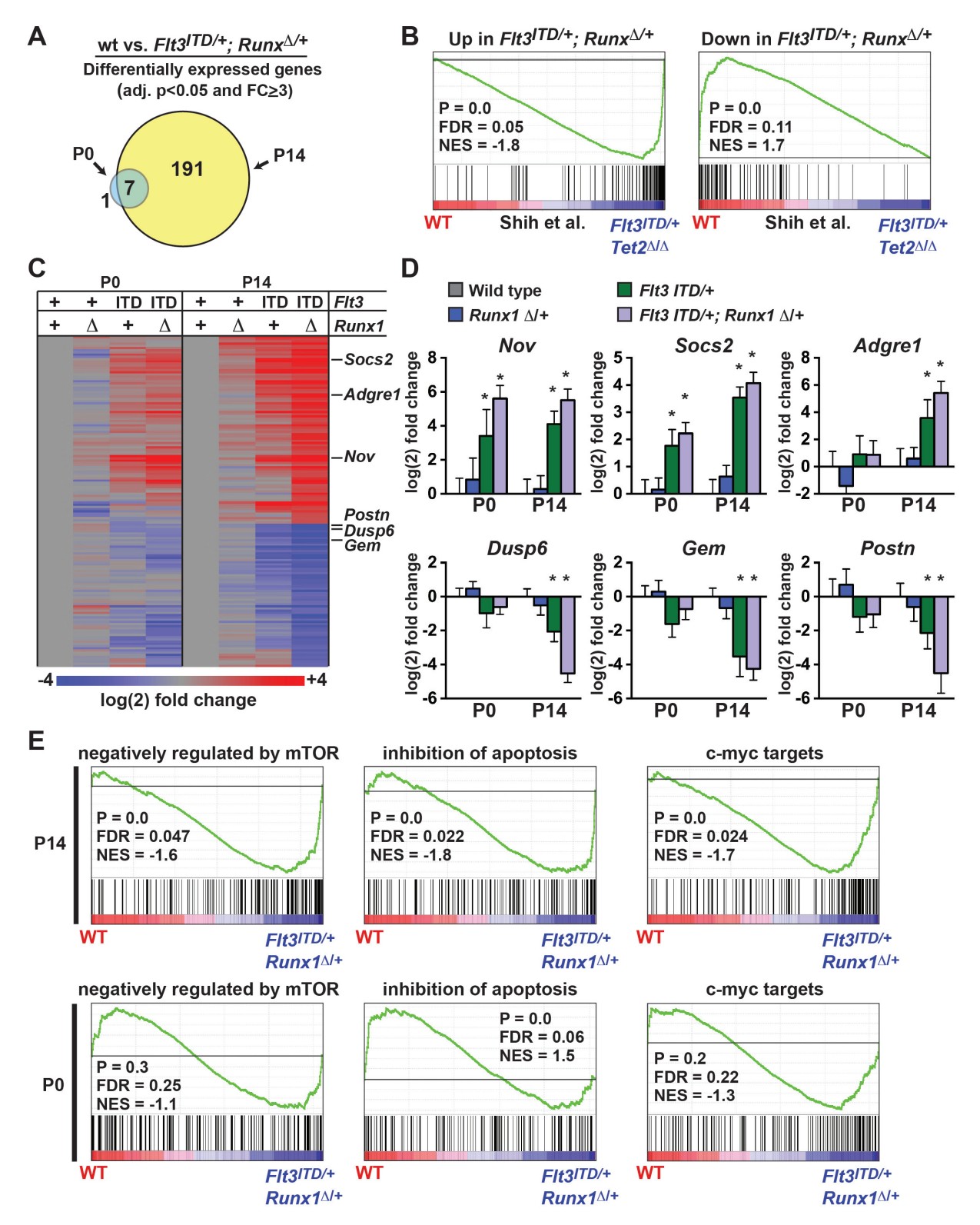

**Figure 9.** *Flt3^ITD* and *Runx1* mutations cooperatively induce changes in gene expression at P14, yet they have a much smaller effect at P0. (**A**) Venn diagram showing overlap between genes that were significantly differentially expressed (adj. p<0.05, fold change ≥ 3) in *Flt3^ITD/+; Runx1^Δ/+* HPCs relative to wild type at P14 or P0; n = 4–5 arrays per genotype and age. (**B**) GSEA shows that differentially expressed genes in *Flt3^ITD/+; Runx1^Δ/+* HPCs overlap significantly with genes that are differentially expressed in *Flt3^ITD/+; Tet2^Δ/Δ* HPCs (*Shih et al., 2015*). (**C**) Heatmap showing expression of genes
*Figure 9 continued on next page*

*Figure 9 continued*

that were differentially expressed in *Flt3^ITD/+^; Runx1^Δ/+^* HPCs relative to wild type HPCs. Each column indicates the average fold change relative to the wild type samples from the same time point. The gene list is shown in **Figure 9—source data1** attached to this figure. (D) Representative examples of genes that are among the most differentially expressed in *Flt3^ITD/+^; Runx1^Δ/+^* HPCs relative to wild type HPCs at P14. Most show much smaller changes in expression at P0. Error bars reflect standard deviations, * adj. p<0.05. (E) GSEA identified several gene sets that were enriched in *Flt3^ITD/+^; Runx1^Δ/+^* HPCs relative to wild type HPCs at P14. Three of the most significantly enriched gene sets are shown for P14 and P0.

The following source data is available for figure 9:

**Source data 1.** *Flt3^ITD^* and *Runx1* mutations cooperatively induce changes in gene expression in post-natal HPCs.

FLT3^ITD^ ectopically activated STAT5-dependent self-renewal programs in HPCs (**Figure 7**). These unanticipated findings suggest that FLT3^ITD^ activates both STAT5-dependent and STAT5-independent signal transduction pathways and that these pathways have opposing effects on self-renewal and myeloid commitment. In this model, STAT5-independent myeloid commitment programs outweigh STAT5-dependent self-renewal programs in the *Flt3^ITD^* mutant bone marrow so the HSC pool becomes depleted (**Figure 6I** and **Figure 7**). *Stat5a/b* deletion can shift the balance further in favor of differentiation, though cooperating mutations may ultimately allow STAT5-dependent self-renewal programs to predominate in transformed AML cells.

The STAT5-independent pathways that antagonize HSC self-renewal and promote myeloid progenitor expansion remain unclear. While the MAPK pathway was hyper-activated in postnatal *Flt3^ITD/+^* HSCs and HPCs, MEK inhibition did not prevent, or even reduce, HSC depletion or HPC expansion in these mice (**Figure 5**). Other candidate pathways, including STAT3 and PI3K, were not activated by FLT3^ITD^ (**Figure 4A**). It is possible that low levels of signal transduction via these pathways were undetectable by Western blot but nevertheless functionally important. It is also possible that an un-interrogated pathway, such as NF-κB or CDK1 (**Gerloff et al., 2015**; **Radomska et al., 2012**), could promote myeloid commitment and antagonize STAT5. Our GSEA data did show increased expression of inflammatory cytokine receptors in *Stat5a/b*-deficient, *Flt3^ITD^* HPCs. This raises the intriguing possibility that inflammatory cytokines could promote differentiation of *Flt3^ITD^* mutant progenitors, and perhaps AML cells. Additional genetic studies are needed to resolve whether these transcriptional changes are a cause or a consequence of enhanced lineage commitment in *Flt3^ITD^* mutant HPCs.

## Developmental programming, re-programming and the origins of pediatric and adult malignancies

Heterochronic genes have been implicated in cancer pathogenesis (**Shyh-Chang and Daley, 2013**). For example, hepatoblastomas, Wilm's tumors and most neuroblastomas present early in life, and they often express the oncofetal proteins LIN28 or LIN28B at high levels to help maintain the primitive differentiation states of their respective anlagen (**Diskin et al., 2012**; **Molenaar et al., 2012**; **Nguyen et al., 2014**; **Urbach et al., 2014**). Adult hepatocellular carcinomas, germ cell tumors and ovarian carcinomas often ectopically activate LIN28 or LIN28B to restore oncofetal programs (**Viswanathan et al., 2009**). MLL-rearranged leukemias have similarly been shown to express embryonic stem cell-related genes (**Somervaille et al., 2009**), and BRAF^V600E^ driven melanoma was recently shown to arise from melanocytes that first de-differentiate into primitive neural crest progenitors (**Kaufman et al., 2016**). In each of these cases, it is easy to appreciate why maintaining or restoring a primitive cell identity might accelerate transformation—fetal cells can proliferate rapidly without differentiating or senescing. However, it is then curious as to why pediatric malignancies are relatively uncommon. Does this simply reflect greater fidelity of the genome at early ages, or are other factors at work?

Our data suggest that the transcriptional regulatory programs of fetal progenitors may, in fact, be protective against some mechanisms of transformation. Fetal and adult progenitors interpret FLT3^ITD^-derived signals differently, as evidenced by their distinct transcriptional responses (**Figures 7** and **8**), and this constrains the ability of FLT3^ITD^ to transform fetal progenitors. Either they lack key transcriptional co-activators, or the epigenetic landscape of fetal progenitors suppresses FLT3^ITD^ target gene activation. Further work is needed to understand the cis- and trans-regulatory elements

that determine when and how individual mutations are competent to transform. If we can understand how normal developmental programs interact with genetic mutations to cause malignancies, it may be possible to target these interactions therapeutically.

## Materials and methods

### Mouse strains

The *Flt3^ITD* RRID:IMSR_JAX:011112 (*Lee et al., 2007*), *Runx1^f*RRID:IMSR_JAX:008772 (*Taniuchi et al., 2002*), *Stat5a/b^f*RRID:MMRRC_032053-JAX (*Cui et al., 2004*), *Rictor^f* RRID:IMSR_ JAX:020649 (*Magee et al., 2012*), *Vav1-Cre* RRID:IMSR_JAX:008610 (*Siegemund et al., 2015*) and *Mx1-Cre* RRID:IMSR_JAX:003556 (*Kühn et al., 1995*) mouse strains have all been previously described and were obtained from The Jackson Laboratory (Bar Harbor, ME). These lines were all on a pure C57BL/6 background. Expression of *Mx1-Cre* was induced by three intraperitoneal injections of pIpC (GE Life Sciences, Pittsburgh, PA; 10 µg/dose) over five days beginning six weeks after birth. PD0325901 (Cayman Chemicals, Ann Arbor, MI) was suspended in 0.5% hydroxypropylmethylcellulose vehicle (Sigma) and administered by oral gavage as described in the text. All mice were housed in the Department for Comparative Medicine at Washington University. All animal procedures were approved by the Washington University Committees on the Use and Care of Animals.

### Isolation of HSCs and flow cytometry

Bone marrow cells were obtained by flushing the long bones (tibias and femurs) or by crushing long bones, pelvic bones and vertebrae with a mortar and pestle in calcium and magnesium-free Hank's buffered salt solution (HBSS), supplemented with 2% heat inactivated bovine serum (Gibco, Carlsbad, CA). Splenocytes were obtained by macerating spleens with frosted slides. Single cell suspensions were filtered through a 40 um cell strainer (Fisher, Houston, TX). The cells were then stained for 20 min with fluorescently conjugated antibodies, washed with HBSS + 2% bovine serum and resuspended for analysis. Cell counts were measured by hemocytometer. The following antibodies were used for flow cytometry, all were from Biolegend (San Diego, CA) except as indicated: CD150 (TC15-12F12.2), CD48 (HM48-1), Sca1 (D7), c-Kit (2B8), Ter119 (Ter-119), CD3 (17A2), CD11b (M1/70), Gr-1 (RB6-8C5), B220 (RA3-6B2), CD8a (53–6.7), CD34 (eBioscience/Affymetrix, Santa Clara, CA, RAM34), CD2 (RM2-5), CD45.1 (A20), CD45.2 (104), CD127 (A7R340), CD16/32 (93) and FLT3/CD135 (A2F10). Lineage stains for all experiments included CD2, CD3, CD8a, Ter119, B220 and Gr1. Antibodies to CD4 and CD11b were omitted from the lineage stains because they are expressed on fetal HSCs at low levels. Unless otherwise indicated, we used the following surface marker phenotypes to define cell populations: HSCs (CD150$^+$, CD48$^-$Lineage$^-$, Sca1$^+$, c-kit$^+$), HPCs (CD48$^-$Lineage$^-$, Sca1$^+$, c-kit$^+$), and GMPs (Lineage$^-$, Sca1$^-$, CD127$^-$, c-kit$^+$, CD34$^+$, CD16/32$^+$). Nonviable cells were excluded from analyses by 4',6-diamidino-2-phenylindone (DAPI) staining (1 µg/ml). When HSCs and HPCs were isolated for Western blotting or RNA collection, c-kit$^+$ cells were enriched prior to sorting by selection with paramagnetic beads (Miltenyi Biotec, Auburn, CA). Flow cytometry was performed on a BD FACSAria Fusion flow cytometer (BD Biosciences).

### Limiting dilution long-term repopulation assays

Eight to ten week old C57BL/6Ka-Thy-1.2 (CD45.1) recipient mice were given two doses of 550 rad delivered at least 3 hr apart. Donor fetal liver or bone marrow cells were mixed with competitor bone marrow cells at the doses indicated in the text and injected via the retroorbital sinus. To assess donor chimerism, peripheral blood was obtained from the submandibular veins of recipient mice at the indicated times after transplantation. Blood was subjected to ammonium-chloride lysis of the red blood cells and leukocytes were stained with antibodies to CD45.2, CD45.1, B220, CD3, CD11b and Gr-1 to assess multilineage engraftment. Functional HSC frequencies were calculated and compared by using Extreme Limiting Dilution Analysis (*Hu and Smyth, 2009*). For secondary transplants, mice were injected with 3 million cells from the bone marrow of primary recipient mice.

### Quantitative RT-PCR

RNA was isolated from HSCs with RNAeasy micro plus columns (Qiagen) and converted to cDNA with Superscript III reverse transcriptase (Lifetech, Carlsbad, CA). Quantitative RT-PCR assays were

performed with Taqman Gene Expression Assays specific to mouse *Flt3* and β-actin (Lifetech). Analysis was performed with a Mx3005P qPCR system (Agilent, Wilmington, DE). Samples were normalized based on β-actin expression.

## Western blots

Twenty-five thousand HSC/MPPs, HPCs or GMPs were double sorted into Trichloracetic acid (TCA), and the volume was adjusted to a final concentration of 10% TCA. Extracts were incubated for 15 min on ice and centrifuged at 16,100xg at 4°C for 10 min. Precipitates were washed in acetone twice and dried. The pellets were solubilized in 9M urea, 2% Triton X-100, 1% DTT. LDS loading buffer (Lifetech) was added and the pellet was heated at 70°C for 10 min. Samples were separated on Bis-Tris polyacrylamide gels (Lifetech) and transferred to a PVDF membrane (Lifetech). All antibodies were from Cell Signaling Technologies (Danvers, MA) except as indicated: P-STAT5 (4322), Total STAT5 (9363), P-STAT3 (9145), Total STAT3 (9139), P-ERK1/2 (4370), Total ERK1/2 (4696), P-AKT Ser473 (4060), P-AKT T308 (13038), Total AKT (4691), α-TUBULIN (3873), β-ACTIN (Santa Cruz Bioscience, Santa Cruz, CA, clone AC-17), HRP-anti-Rabbit IgG (7074) and HRP-anti-mouse IgG (7076). Blots were developed with the SuperSignal West Femto or Pico chemiluminescence kits (Thermo Scientific). Blots were stripped (1% SDS, 25 mM glycine pH 2) prior to re-probing.

## Cytospins

Bone marrow cells were isolated and spun onto glass slides using a Shandon Cytospin 3. The slides were stained using Protocol Hema 3 Wright-Giemsa stain (Fisher Scientific). All slides were reviewed by a pediatric hematologist (JAM or ASC).

## Statistical analysis

In all cases, multiple independent experiments were performed on at least three different days to verify that the data are reproducible. Grouped data reflect biological replicates (i.e. independent mice) and are represented by mean +/− standard deviation. Statistical comparisons between groups were made with the two-tailed Student's t-test except as noted in the figure legends. When multiple genotypes were compared, statistical significance was determined by performing a one-way ANOVA followed by a Holm-Sidak post-hoc test to correct for multiple comparisons. For transplantation experiments, the percentages of mice with multilineage reconstitution were compared with the Fisher's exact test. All comparisons were performed with GraphPad Prism 6, RRID:SCR_002798.

## Gene expression analysis

Ten thousand HSCs or HPCs were double sorted directly into RLT plus RNA lysis buffer (Qiagen) and RNA was isolated with RNAeasy micro plus columns (Qiagen). Transcripts were amplified with the WTA2 kit (Sigma) with the Kreatech ULS RNA labeling kit (Kreatech Diagnostics). Labeled cDNA was hybridized to Agilent Mouse 8 × 60 K microarrays and analyzed with an Agilent C-class scanner. Signal data were assembled and processed in Partek RRID:SCR_011860, and samples were compared by Linear Models for Microarrays, RRID:SCR_010943 (*Ritchie et al., 2015*; *Smyth, 2004*). Adjusted p-values were calculated by the Benjamini and Hochberg false discovery rate (*Benjamini and Hochberg, 1995*). Z-scores were calculated as previously described (*Cheadle et al., 2003*). Hierarchical cluster analysis was performed with Cluster 3.0 and visualized with Java TreeView; RRID:SCR_013505 and RRID:SCR_013503 (*Eisen et al., 1998*). Principal component analyses and Euclidean distance comparisons (by permutation testing) were performed with the R software environment. Microarray data sets have been deposited into Gene Expression Omnibus (GSE81153). GSEA was performed using gene sets that were generated as cited in the text, or with gene sets curated in the MSigDB databases; RRID:SCR_003199 (*Subramanian et al., 2005*).

## Acknowledgements

This work was supported by grants from the Department of Defense (CA130124), the St. Baldrick's Foundation, Hyundai Hope on Wheels, the Gabrielle's Angel Foundation for Cancer Research and the Children's Discovery Institute of Washington University and St. Louis Children's Hospital. JAM is a scholar of the Child Health Research Center for Excellence in Developmental Biology at

Washington University (K12-HD076224). ASC is supported by a training grant to the Washington University Department of Pediatrics (5T32HD043010-12). We thank Jenna Voigtmann for technical assistance. We thank D Bhattacharya, G Challen, I Maillard and R Signer for comments on the manuscript.

## Additional information

### Funding

| Funder | Grant reference number | Author |
|---|---|---|
| Eunice Kennedy Shriver National Institute of Child Health and Human Development | 5T32HD043010-12 | Andrew S Cluster |
| U.S. Department of Defense | CA130124 | Jeffrey A Magee |
| St. Baldrick's Foundation | Scholar Award | Jeffrey A Magee |
| Hyundai Hope On Wheels | Hope Scholar | Jeffrey A Magee |
| Gabrielle's Angel Foundation for Cancer Research | Medical Research Award | Jeffrey A Magee |
| Children's Discovery Institute of Washington University and St. Louis Children's Hospital | Faculty Scholar Award | Jeffrey A Magee |
| Eunice Kennedy Shriver National Institute of Child Health and Human Development | K12-HD076224 | Jeffrey A Magee |

The funders had no role in study design, data collection and interpretation, or the decision to submit the work for publication.

### Author contributions

SNP, JAM, Conception and design, Acquisition of data, Analysis and interpretation of data, Drafting or revising the article; ASC, KAB, RMP, JR, Acquisition of data, Analysis and interpretation of data; WY, Conception and design, Analysis and interpretation of data, Drafting or revising the article

### Author ORCIDs

Jeffrey A Magee, http://orcid.org/0000-0002-0766-4200

### Ethics

Animal experimentation: All mice were housed in the Department for Comparative Medicine at Washington University. All animals were handled and procedures were performed according to institutional animal care and use committee (IACUC) protocols 20130134 and 20160087. These protocols were approved by the Washington University Committees on the Use and Care of Animals.

## Additional files

### Major datasets

The following dataset was generated:

| Author(s) | Year | Dataset title | Dataset URL | Database, license, and accessibility information |
|---|---|---|---|---|
| Jeffrey M | 2016 | Fetal and neonatal hematopoietic progenitors are functionally and transcriptionally resistant to Flt3-ITD mutations | https://www.ncbi.nlm.nih.gov/geo/query/acc.cgi?acc=GSE81153 | Publicly available at the NCBI Gene Expression Omnibus (accession no: GSE81153) |

The following previously published dataset was used:

| Author(s) | Year | Dataset title | Dataset URL | Database, license, and accessibility information |
|---|---|---|---|---|
| Levine RL, Shin A | 2015 | Tet2-/-Flt3ITD and WT stem and progenitor cells | http://www.ncbi.nlm.nih.gov/geo/query/acc.cgi?acc=GSE57244 | Publicly available at the NCBI Gene Expression Omnibus (accession no: GSE57244) |

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
