## [Decision Letter]

Thank you for submitting your article "Fetal and neonatal hematopoietic progenitors are functionally and transcriptionally resistant to *Flt3*-ITD mutations" for consideration by *eLife*. Your article has been reviewed by two peer reviewers, and the evaluation has been overseen by a Reviewing Editor and Charles Sawyers as the Senior Editor. The reviewers have opted to remain anonymous.

The reviewers have discussed the reviews with one another and the Reviewing Editor has drafted this decision to help you prepare a revised submission.

Summary:

In the manuscript by Porter et al., the authors investigate the effects of FLT3^ITD^ on hematopoietic stem and progenitor cells across the developmental timeline from fetal to neonatal to adult mice. They find that FLT3^ITD^ is able to impair HSC self-renewal and expand progenitors leading to MPN only in adult mice, and specifically not in fetal or neonatal mice. They use gene expression profiling to show that while STAT5 signaling is active in fetal FLT3^ITD^ cells, these cells do not express the gene expression programs associated with the adult phenotypes. They argue that fetal and neonatal progenitors may be protected from leukemic transformation because they are not competent to express FLT3^ITD^ target genes, and that this may help explain why FLT3^ITD^ is not commonly observed in pediatric AML. In general, this question of oncogene/driver mutation differences between adult and pediatric AML is interesting and relevant. This is well done and novel and of interest to the field. However, there are a number of issues that must be addressed:

Essential revisions:

1) Can the authors use flow to support the mRNA studies of FLT3 expression in different stem/progenitor cells?

2) Why are HSCs lost in BM but not in spleen with the ITD? Are spleen HSCs with ITD functionally competent? This should be addressed ideally with further experiments but at least a discussion as to the possible reasons for the difference.

3) Are there differences in transplantability of the stem cell or disease phenotype of ITD/*runx1* AML cells based on fetal versus adult origin? Some more detailed characterization of the phenotypes would be of interest.

4) The STAT5 results are very interesting and surprising. A few questions emerge. Is this due to loss of both total and unphosphorylated STAT5, as Tony Green has shown that STAT5 can bind different targets based on whether it is or is not phosphorylated? Is it due to differences or antagonism between *Stat5a* and *b*?

5) The authors propose a provocative model (Figure 6), in which FLT3^ITD^ acts in a STAT5-dependent manner to promote HSC self-renewal, and in a STAT5-independent manner to promote myeloid commitment/differentiation. This paper would be strengthened by inclusion of data supporting this hypothesis. In particular, the gene expression data could be analyzed to interrogate this model, as STAT5-dependent and STAT5-independent FLT3^ITD^ target genes were identified in Figure 7. Are the STAT5-dependent target genes enriched for genes involved in HSC self-renewal? Are the STAT5-independent target genes enriched for genes involved in myeloid commitment/differentiation? GSEA could be used to investigate these questions. Moreover, do these genes provide any clues as to what pathways might be mediating these effects, particularly the STAT5-independent effects?

6) The authors note that there were no significant gene expression differences between FLT3^ITD^ and wild type HSCs at any of the time points investigated. This is surprising given the markedly different phenotypes associated with HSC functions at these time points. The low level of expression of FLT3 cannot explain this finding given that the HSC have a striking phenotype, even with this low level of expression. This issue requires further investigation/explanation.

7) The description of the phenotypes of fetal, neonatal, and adult FLT3^ITD^ HSC/HPC are well-conducted and clearly support the conclusion that FLT3^ITD^ affects these cells differently. However, the additional studies to investigate the mechanism of these differences need some clarification. First, the inhibition of MAPK signaling did not prevent the HSC deletion, HPC expansion, or MPN observed in the adult mice. Second, and most strikingly, the STAT5 deletion led to an exacerbated phenotype rather than rescue. In the Discussion, the authors correctly highlight several possible mechanisms: "lack key transcriptional co-activators, or the epigenetic landscape of fetal progenitors suppresses FLT3^ITD^ target gene activation". This paper would be greatly strengthened by any mechanistic data at least partially explaining these findings.

---

## [Author Response]

*Essential revisions:*

*1) Can the authors use flow to support the mRNA studies of FLT3 expression in different stem/progenitor cells?*

We have performed flow cytometry to assess FLT3 expression in fetal and adult progenitors. FLT3 was expressed in hematopoietic stem/progenitor cells in both E14.5 and 8 week old mice, consistent with the mRNA expression data. These data are added as a new panel to Figure 1 (Figure 1).

*2) Why are HSCs lost in BM but not in spleen with the ITD? Are spleen HSCs with ITD functionally competent? This should be addressed ideally with further experiments but at least a discussion as to the possible reasons for the difference.*

The purpose of evaluating spleen HSC frequencies in *Flt3^ITD/ITD^* mice (Figure 1 in the current draft) was to demonstrate that depletion of HSCs in the bone marrow was not accompanied by a simultaneous expansion of HSCs in the spleen. This contrasts with the effect of other leukemogenic mutations that deplete the bone marrow HSC pool, particularly *Pten* loss of function (Nature 441:475, Stem Cell Reports 6:806). The frequency of HSCs in the adult wild type spleen is so low that it would be difficult to reproducibly detect further depletion in *Flt3^ITD/ITD^* mice. A better assessment can be made at P14, when phenotypic HSCs are present in the wild type spleen at frequencies similar to the bone marrow. As shown in Figure 3, *Flt3^ITD^* causes HSC depletion and HPC expansion in both the bone marrow and the spleen. We have added text to illuminate this point and to provide a more clear rationale for Figure 1.

*3) Are there differences in transplantability of the stem cell or disease phenotype of ITD/runx1 AML cells based on fetal versus adult origin? Some more detailed characterization of the phenotypes would be of interest.*

We competitively transplanted P0 liver cells and P21 bone marrow cells from control and *Flt3^ITD^; Runx1* compound mutant mice (N=15 recipients from 3 independent donors per age and genotype). At four weeks after transplantation, all 15 recipients of P0 *Flt3^ITD^; Runx1* mutant donor cells were multi-lineage reconstituted. In contrast, only 1 of 15 recipients of P21 *Flt3^ITD^; Runx1* mutant donor cells was multi-lineage reconstituted. The different repopulating activities of P0 and P21 *Flt3^ITD^; Runx1* mutant donor cells were evident even when we looked specifically at myeloid reconstitution, indicating that these differences are not simply a consequence of altered lineage bias in the compound mutant progenitors. These new data show that cooperating FLT3^ITD^ and *Runx1* mutations have developmental context-specific effects on the functional properties of HSCs and HPCs. These data have been added as new panels M-P in Figure 3.

*4) The STAT5 results are very interesting and surprising. A few questions emerge. Is this due to loss of both total and unphosphorylated STAT5, as Tony Green has shown that STAT5 can bind different targets based on whether it is or is not phosphorylated? Is it due to differences or antagonism between Stat5a and b?*

The *Stat5a/b* conditional loss-of-function mice used in this study have loxP sites flanking both *Stat5a* and *Stat5b*. Thus, the changes that we have observed are due to a complete loss of STAT5 protein rather than changes in the relative levels of *Stat5a* and *Stat5b* or changes in the ratio of phosphorylated to unphosphorylated protein. We have added a Western blot confirming that both phosphorylated and unphosphorylated STAT5 are eliminated in HPCs after conditional *Stat5a/b* deletion. These data are in Figure 6—figure supplement 1.

*5) The authors propose a provocative model (Figure 6), in which* FLT3^ITD^*acts in a STAT5-dependent manner to promote HSC self-renewal, and in a STAT5-independent manner to promote myeloid commitment/differentiation. This paper would be strengthened by inclusion of data supporting this hypothesis. In particular, the gene expression data could be analyzed to interrogate this model, as STAT5-dependent and STAT5-independent* FLT3^ITD^*target genes were identified in Figure 7. Are the STAT5-dependent target genes enriched for genes involved in HSC self-renewal? Are the STAT5-independent target genes enriched for genes involved in myeloid commitment/differentiation? GSEA could be used to investigate these questions. Moreover, do these genes provide any clues as to what pathways might be mediating these effects, particularly the STAT5-independent effects?*

To address this question, we generated self-renewal-related and commitment- related gene sets from our analyses of wild type HSCs and HPCs. The self-renewal-related gene set consisted of genes that were expressed at significantly higher levels in HSCs as compared to HPCs (>5 fold, adj. p<0.05). The commitment-related gene set consisted of genes that were expressed at significantly higher levels in HPCs as compared to HSCs. We then used GSEA to compare HPCs from wild type, *Flt3^ITD^* and *Flt3^ITD^; Stat5a/b* compound mutant mice. The self-renewal gene set was significantly enriched in *Flt3^ITD/+^*HPCs as compared to wild type HPCs, and the commitment-related gene set was significantly enriched in wild type HPCs as compared to *Flt3^ITD/+^* HPCs. These enrichment patterns were reversed when *Stat5a/b* was deleted. We next looked at an independently curated self-renewal gene set (Ivanova et al., Science 298:601). This gene set was significantly enriched in wild type and *Flt3^ITD^* HPCs relative *to Flt3^ITD^; Stat5a/b* compound mutant HPCs. These observations show that *Flt3* ectopically activates self-renewal-related programs in HPCs via STAT5-dependent mechanisms. Furthermore, *Flt3^ITD^* promotes myeloid commitment via STAT5-independent mechanisms. These observations agree with the model proposed in Figure 6, and they have been added as part of a new Figure 7 (panels C and D).

To identify potential STAT5-independent mechanisms that promote myeloid commitment, we performed GSEA using gene sets curated in the MSigDB database. Several of the most significantly enriched gene sets reflected increased inflammatory cytokine signaling (e.g. Interferon signaling). This raises the possibility that inflammatory cytokines may contribute to myeloid differentiation in *Flt3^ITD^* HSCs/HPCs. These data have been added as part of a new Figure 7 (panel E). Future experiments will test whether these changes in cytokine signaling are a cause or a consequence of increased differentiation in *Flt3^ITD^* mutant HPCs.

*6) The authors note that there were no significant gene expression differences between* FLT3^ITD^*and wild type HSCs at any of the time points investigated. This is surprising given the markedly different phenotypes associated with HSC functions at these time points. The low level of expression of FLT3 cannot explain this finding given that the HSC have a striking phenotype, even with this low level of expression. This issue requires further investigation/explanation.*

We agree that this outcome was surprising, but note that the criteria used to identify significant changes in gene expression were stringent with the intention of excluding false positive hits from the screen. Many of the genes that were identified as being differentially expressed in *Flt3^ITD/+^* HPCs are also differentially expressed in HSCs (P<0.05) when each gene is interrogated individually without correction for multiple hypothesis testing. This suggests that FLT3^ITD^ does induce changes in gene expression in HSCs, but the measured effects are modest relative to HPCs. These analyses have been added to [Supplementary-material SD3-data].

One potential explanation for the different effects seen in HSCs and HPCs is that strong activation of FLT3^ITD^ target genes may cause HSCs to differentiate, at least partially, into HPCs. Thus, our arrays may have only captured the most mildly affected HSCs. This possibility is consistent with our model showing that FLT3^ITD^ induces both self-renewal and lineage commitment programs. This explanation has been added to the text.

*7) The description of the phenotypes of fetal, neonatal, and adult* FLT3^ITD^*HSC/HPC are well-conducted and clearly support the conclusion that* FLT3^ITD^*affects these cells differently. However, the additional studies to investigate the mechanism of these differences need some clarification. First, the inhibition of MAPK signaling did not prevent the HSC deletion, HPC expansion, or MPN observed in the adult mice. Second, and most strikingly, the STAT5 deletion led to an exacerbated phenotype rather than rescue. In the Discussion, the authors correctly highlight several possible mechanisms: "lack key transcriptional co-activators, or the epigenetic landscape of fetal progenitors suppresses* FLT3^ITD^*target gene activation". This paper would be greatly strengthened by any mechanistic data at least partially explaining these findings.*

We have added several pieces of additional data to Figure 7 to address the potential mechanisms by which *Flt3^ITD^*and STAT5 regulate HSC and HPC fate. These findings are described above.

We agree with the reviewers that the next important step will be to define the cis-regulatory elements and transcription factors responsible for the temporal changes in FLT3^ITD^ target gene regulation. This is a very large undertaking that we feel is beyond the scope of this current manuscript. The assays that we are performing to address these mechanistic questions (single cell RNA-seq, ATAC-seq, ChIP-seq and characterization of compound mutant mice) will take many months if not years to complete. In our opinion, the key points in this paper are novel and well supported by the data that we have provided. We hope that the reviewers will agree that additional mechanistic insights will be most effectively handled in a follow-up paper.